# 'Artiphysiology' reveals V4-like shape tuning in a deep network trained for image classification

Dean A Pospisil[1]*, Anitha Pasupathy[1,2], Wyeth Bair[1,2,3]

[1]Department of Biological Structure, Washington National Primate Research Center, University of Washington, Seattle, United States; [2]University of Washington Institute for Neuroengineering, Seattle, United States; [3]Computational Neuroscience Center, University of Washington, Seattle, United States

**Abstract** Deep networks provide a potentially rich interconnection between neuroscientific and artificial approaches to understanding visual intelligence, but the relationship between artificial and neural representations of complex visual form has not been elucidated at the level of single-unit selectivity. Taking the approach of an electrophysiologist to characterizing single CNN units, we found many units exhibit translation-invariant boundary curvature selectivity approaching that of exemplar neurons in the primate mid-level visual area V4. For some V4-like units, particularly in middle layers, the natural images that drove them best were qualitatively consistent with selectivity for object boundaries. Our results identify a novel image-computable model for V4 boundary curvature selectivity and suggest that such a representation may begin to emerge within an artificial network trained for image categorization, even though boundary information was not provided during training. This raises the possibility that single-unit selectivity in CNNs will become a guide for understanding sensory cortex.

*For correspondence:
deanp3@uw.edu

Competing interests: The authors declare that no competing interests exist.

## Introduction

Deep convolutional neural networks (CNNs) are currently the highest performing image recognition computer algorithms. While their overall design reflects the hierarchical structure of the ventral ('form-processing') visual stream (*Hubel and Wiesel, 1962*; *LeCun et al., 2015*), the visual selectivity (i.e., tuning) of single units within the network are not constrained to match neurobiology. Rather, single-unit properties are determined by a performance-based learning algorithm that operates iteratively across many pre-classified training images, tuning the parameters of the network to decrease the error between the network output and the target classification. Nevertheless, first-layer units in these CNNs, following training, often show selectivity for orientation and spatial frequency (*Figure 1*; see also *Krizhevsky et al., 2012*) like neurons in primary visual cortex (V1). Attempts to visualize features encoded by single units deeper in such networks (*Zeiler and Fergus, 2013*; *Mahendran and Vedaldi, 2014*) show that selectivity becomes increasingly complex and categorical, similar to the progression along the ventral stream. Solidifying this idea, *Güçlü and van Gerven, 2015* found a corresponding hierarchy of visual features between BOLD signals in the human ventral stream and layers within a CNN. This raises the tentative but exciting possibility that units deeper in the network may approximate tuning observed at mid-level stages of the ventral stream, for example area V4. This is not unreasonable given that artificial networks that perform better at image classification also have population-level representations closer to those in area IT (*Yamins et al., 2014*; *Khaligh-Razavi and Kriegeskorte, 2014*; *Kriegeskorte, 2015*). V4 is a primary input to IT (*Felleman and Van Essen, 1991*), yet there has been no systematic examination of whether specific form-selective properties found in V4 emerge within a CNN.

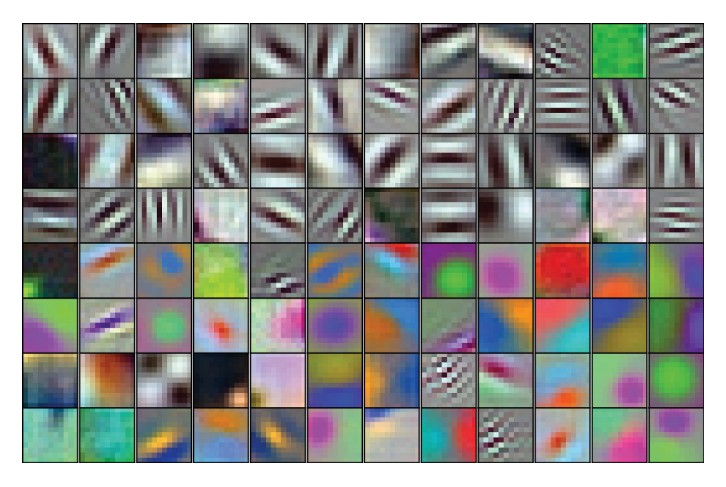

**Figure 1.** The 96 kernels (11 × 11 pixels, by three color channels) of the 1 st layer, Conv1, of the AlexNet model tested here. Like many V1 receptive fields, many of these kernels are band-limited in spatial frequency and orientation. Each kernel was independently scaled to maximize its RGB dynamic range to highlight spatial structure.

To address this, we tested whether two properties of shape selectivity in V4, tuning for boundary curvature (*Pasupathy and Connor, 1999*; *Pasupathy and Connor, 2001*; *Cadieu et al., 2007*) and translation invariance (*Gallant et al., 1996*; *Pasupathy and Connor, 2001*; *Rust and Dicarlo, 2010*; *Rust and DiCarlo, 2012*; *Nandy et al., 2013*; *Sharpee et al., 2013*) arise within a CNN. In particular, many V4 neurons are selective for boundary curvature, ranging from concave to sharply convex, at particular angular positions around the center of an object. This angular position and curvature (APC) tuning may be important for supporting entire object representations deeper in the ventral stream (*Pasupathy and Connor, 2001*; *Murphy and Finkel, 2007*), but it remains uncertain how it arises or is used. Finding APC-like tuning in the middle of an artificial network could help to relate mid-level visual physiology to pressures on visual representation applied by image statistics at the front end and by categorization performance downstream. It could also relate to the recent observation that human perception of shape similarity correlates with response similarity in CNNs (*Kubilius et al., 2016*).

We take an approach to characterizing single units in an artificial deep network that we refer to as 'artiphysiology' because it closely mirrors how an electrophysiologist approaches the characterization of single neurons in the brain. In particular, we presented the original 362 shape stimuli used by *Pasupathy and Connor, 2001* to AlexNet, a CNN that was the first of its class to make large gains on general object recognition (*Krizhevsky et al., 2012*) and that continues to be well-studied (*Zeiler and Fergus, 2013*; *Yosinski et al., 2015*; *Lenc and Vedaldi, 2014*; *Donahue et al., 2014*; *Szegedy et al., 2013*; *Güçlü and van Gerven, 2015*; *Bau et al., 2017*; *Tang et al., 2017*; *Flachot and Gegenfurtner, 2018*). Making direct comparisons between CNN units and V4 neurons using V4 data from two previous studies (*Pasupathy and Connor, 2001*; *El-Shamayleh and Pasupathy, 2016*), we found that many units in AlexNet would be indistinguishable from good examples of boundary-curvature-tuned V4 neurons. We applied a CNN visualization technique (*Zeiler and Fergus, 2013*) to examine whether natural image features that best drive such APC-like units are consistent with the notion of selectivity for curvature of object boundaries. We identify specific V4-like units so that other researchers may utilize them for future studies.

## Results

AlexNet contains over 1.5 million units organized in eight major layers (*Figure 2*), but its convolutional architecture means that the vast majority of those units are spatially offset copies of each other. For example, in the first convolutional layer, Conv1, there are only 96 distinct kernels (*Figure 1*), but they are repeated everywhere on a 55 × 55 grid (*Figure 2E*). Thus, for the convolutional

layers, Conv1 to Conv5, it suffices to study the selectivity of only those units at the spatial center of each layer. These units, plus all units in the subsequent fully-connected layers comprise the 22,096 unique units (*Figure 2D*) that we analyzed.

## Responses of CNN units to simple shapes

We first establish that the simple visual stimuli used in V4 electrophysiology experiments (*Figure 3A*) do in fact drive units within the CNN, which was trained on a substantially different set of inputs: natural photographic images from the ImageNet database (*Deng et al., 2009*). Across the convolutional layers and their sublayers, we found that our shape stimuli typically evoked a range of responses that was on average similar to, or larger than, the range driven by ImageNet images (e.g., *Figure 4*, Conv1, compare red to dark blue). The ranges for shapes and images became more similar following normalization layers (e.g., *Figure 4*, Norm1). In contrast, in the subsequent fully-connected layers, the natural images drove a larger range of responses (*Figure 4*, FC6, dark blue) than did shapes (red line), and from FC6 onwards the range of responses to shapes was about 1/2 to 1/3 of that for images. The wider dynamic range for images in later layers may reflect the sensitivity of deeper units to category-relevant conjunctions of image statistics that are absent in our simple shape stimuli. These results were robust to changes in stimulus intensity and size (see *Figure 4*, legend); therefore, we settled on a standard size of 32 pixels so that stimuli fit within all RFs from Conv2 onwards (*Figure 2B*) with room to spare for translation invariance tests (see Materials and methods).

Although our shapes drove responses in all CNN layers, many units responded sparsely to both the shapes and natural images. Across all layers, 13% of units had zero responses to all shape stimuli and 7% had non-zero response to only one stimulus, that is one shape at one rotation. Because we aim to identify CNN units with V4-like responses to shapes, we excluded from further analysis units with response sparseness outside the range observed in V4 (see Materials and methods and *Figure 4—figure supplement 1*).

## Tuning for boundary curvature at RF center

To assess whether CNN units have V4-like boundary curvature selectivity, we measured responses of each unique CNN unit to our shape stimuli (up to eight rotations for each shape in *Figure 3A*), centered in the RF. We then fit responses with the angular position and curvature (APC) model (*Pasupathy and Connor, 2001*), which captures neuronal selectivity as the product of a Gaussian tuning curve for curvature and a Gaussian tuning curve for angular position with respect to the center of the shape (*Figure 3B,C* and Materials and methods). We found that the responses of many units in the CNN were fit well by the APC model. For example, the responses of Conv2 unit 113 (i. e., Conv2-113) were highly correlated ($r = 0.78$, $n = 362$) to those of its best-fit APC model (*Figure 5A*). The fit parameters indicate selectivity for a sharp convexity ($\mu_c = 1.0$, $\sigma_c = 0.39$) pointing to the upper left ($\mu_a = 135°$, $\sigma_a = 23°$), and indeed the eight most preferred shapes all include such a feature (*Figure 5B*, pink), whereas the least preferred shapes (cyan) do not. A second example unit, FC7-3591 (*Figure 5C*) with a high APC r-value (0.77) had fit parameters (see legend) reflecting selectivity for concavities roughly toward the top of the shape, consistent with most of its preferred shapes (*Figure 5D*). These results were similar to those for well-fit V4 neurons. For example, the V4 unit *a1301* (*Figure 5E,F*) had an APC fit ($r = 0.76$, $p < 0.001$, $n = 362$) reflecting a preference for a sharp convexity, like the first CNN example unit, except with a different preferred angular position ($\mu_a = 180°$).

For each layer of the CNN, we computed the distributions of the APC fit r-values across units (*Figure 5G*). There is a clear but modest trend for the cumulative distribution functions to shift rightward for higher layers (orange lines, *Figure 5G*), indicating that deeper layer units fit better on average to the APC model. The first CNN layer, Conv1 (black line) stands apart as having a far leftward-shifted r-value distribution, but this occurs simply because most of the stimuli overfill the small Conv1 RFs. Compared to V4 neurons studied with the same shape set (red line, *Figure 5G*), the median r-values (corresponding to 0.5 on the vertical axis) for layers Conv2 to FC8 were somewhat higher than that for V4, but the V4 and CNN curves matched closely at the upper range, with the best V4 unit having a higher APC r-value than any CNN unit.

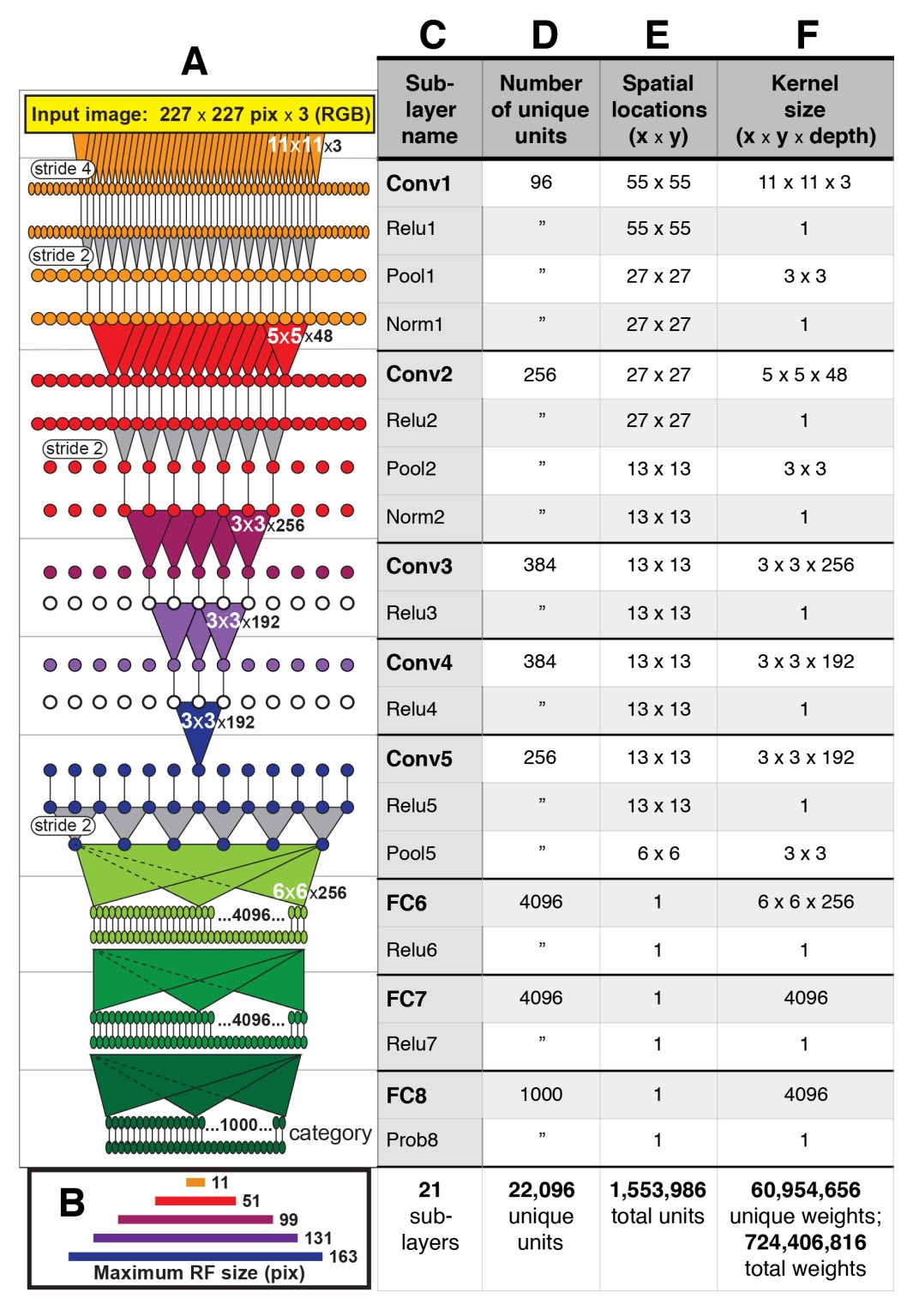

| | Sub-layer name | Number of unique units | Spatial locations (x x y) | Kernel size (x x y x depth) |
|---|---|---|---|---|
| **C** | **D** | **E** | **F** | |
| Conv1 | 96 | 55 x 55 | 11 x 11 x 3 |
| Relu1 | " | 55 x 55 | 1 |
| Pool1 | " | 27 x 27 | 3 x 3 |
| Norm1 | " | 27 x 27 | 1 |
| Conv2 | 256 | 27 x 27 | 5 x 5 x 48 |
| Relu2 | " | 27 x 27 | 1 |
| Pool2 | " | 13 x 13 | 3 x 3 |
| Norm2 | " | 13 x 13 | 1 |
| Conv3 | 384 | 13 x 13 | 3 x 3 x 256 |
| Relu3 | " | 13 x 13 | 1 |
| Conv4 | 384 | 13 x 13 | 3 x 3 x 192 |
| Relu4 | " | 13 x 13 | 1 |
| Conv5 | 256 | 13 x 13 | 3 x 3 x 192 |
| Relu5 | " | 13 x 13 | 1 |
| Pool5 | " | 6 x 6 | 3 x 3 |
| FC6 | 4096 | 1 | 6 x 6 x 256 |
| Relu6 | " | 1 | 1 |
| FC7 | 4096 | 1 | 4096 |
| Relu7 | " | 1 | 1 |
| FC8 | 1000 | 1 | 4096 |
| Prob8 | " | 1 | 1 |
| **21 sub-layers** | **22,096 unique units** | **1,553,986 total units** | **60,954,656 unique weights; 724,406,816 total weights** |

**Figure 2.** Architecture of the Caffe AlexNet CNN. (**A**) A one-dimensional scale view of the fan-in and spatial resolution of units for all 21 sublayers, aligned to their names listed in column (**C**). The color-filled triangles in convolutional (Conv) layers indicate the fan-in to convolutional units, gray triangles indicate the fan-in to max pooling units, and circles (or ovals) indicate the spatial positions of units along the horizontal dimension. For the Conv layers and their sublayers, each circle in the diagram represents the number of unique units listed in column (**D**). For example, for each orange circle/oval in the four sublayers associated with Conv1, there are 96 different

*Figure 2 continued*

units in the model (the Conv1 kernels are depicted in *Figure 1*). The 227 pixel wide input image (top, yellow), is subsampled at the Conv1 sublayer (orange; 'stride 4' indicates that units occur only every four pixels) and again at each pooling sublayer ('stride 2'), until the spatial resolution is reduced to a $6 \times 6$ grid at the transition from Pool5 to FC6. The pyramid of support converging to the central unit in Conv5 (dark blue triangle) is indicated by triangles and line segments starting from Conv1. Each unit in layers FC6, FC7 and FC8 (shades of green; not all units are shown) receives inputs from all units in the previous layer (there is no spatial dimension in the FC layers, units are depicted in a line only for convenience). Green triangles indicate the full fan-in to three example units in each FC layer. (B) The maximum width (in pixels) of the RFs for units in the five convolutional layers (colors match those in (A)) based on fan-in starting from the input image. For the FC layers, the entire image is available to each unit. (C) Names of the sublayers, aligned to the circuit in (A). Names in bold correspond to the eight major layers, each of which begins with a linear kernel (colorful triangles in (A)). (D) The number of unique units, that is feature dimensions, in each sublayer (double quotes repeat values from previous row). (E) The width and height of the spatial (convolutional) grid at each sublayer, or '1' for the FC layers. The total number of units in each sublayer can be computed by multiplying the number of unique kernels (D) by the number of spatial positions (E). (F) The kernel size corresponds to the number of weights learned for each unique linear kernel. Pooling layers have $3 \times 3$ spatial kernels but have no weights—the maximum is taken over the raw inputs. The Conv2 kernels are only 48 deep because half of the Conv2 units take inputs from the first 48 feature dimensions in Conv1, whereas the other half take inputs from the last 48 Conv1 features; inputs are similarly grouped in Conv4 and Conv5 (see Krizhevsky et al.'s *Figure 2*). The bottom row provides totals. In addition to the weights associated with each kernel, there is also one bias value per kernel (not shown), which adds 10,568 free parameters to the ~60.9 million unique weights.

One factor that could influence our CNN to V4 comparison is that CNN responses are noise-free, whereas V4 responses have substantial trial-to-trial variability. We extended the method of *Haefner et al. (2009)* to remove the bias that variability introduces into the correlation coefficient (see Materials and methods). The distribution of the corrected estimates of the r-values across the V4 population (pink line, *Figure 5G*) has a higher median than that for any of the CNN layers. This suggests that, had it been possible to record many more stimulus repeats to eliminate most of the noise in the V4 data, then one would find that the V4 population somewhat out-performs even the deep layers in AlexNet in fitting the APC model. Overall, regardless of whether we consider the raw or corrected V4 r-values, we would still conclude that the CNN contains units that cover the vast majority of the range of APC r-values found in V4 when tested with the same stimuli.

To determine whether the goodness of fit to the APC model was a result of the network architecture alone or if training on the object categorization task played a role, we fit the model to units in an untrained network in which weights were assigned random initial values (see Materials and methods) and found that only ~14% had APC r-values above 0.5 (*Figure 5H*, blue trace) and none reached the upper range of r-values observed in the trained CNN (*Figure 5H*, black line, aggregate of all layers) or in V4 (red line). This suggests that training is important for achieving an APC r-value distribution consistent with V4.

To control for over fitting, we re-fit the APC model to all CNN units after shuffling the responses of each unit across the 362 shapes. After shuffling, 99% of units had $r < 0.07$ (*Figure 5H*, green), whereas in the original data (*Figure 5H*, black) 99% of units had $r > 0.07$. Thus, the APC model largely reflects specific patterns of responses of the units to the shapes, and not an ability of the model to fit any random or noisy set of responses (see also *Pasupathy and Connor, 2001*).

## Translation Invariance

To have V4-like boundary curvature tuning, a CNN unit must not only fit the APC model well for stimuli centered in the RF, but must maintain that selectivity when stimuli are placed elsewhere in the RF, that is it must show translation invariance like that found in V4 for our stimulus set (*Pasupathy and Connor, 2001*; *El-Shamayleh and Pasupathy, 2016*). For example, responses of a V4 neuron to 56 shapes centered in the RF are highly correlated ($r = 0.97, p < 0.0001$, n = 56) with responses to the same shapes presented at a location offset by 1/6 of the RF diameter (*Figure 6A*), indicating that shapes that drive relatively high (or low) responses at one location also tend to do so at the other location. This can be visualized across the RF using the position-correlation function (*Figure 6B*, red), which plots response correlation as a function of distance from a reference position

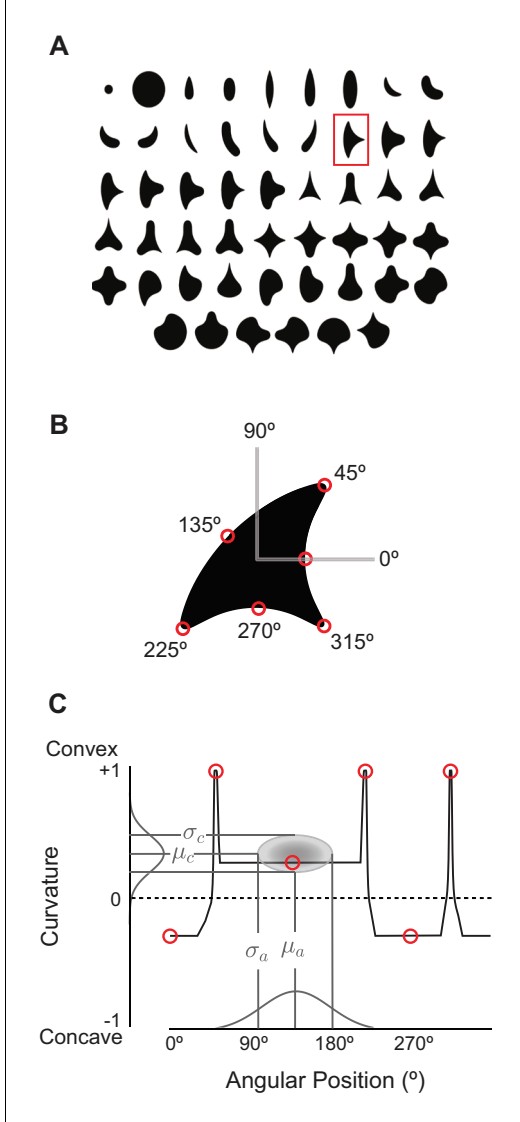

**Figure 3.** The angular position and curvature (APC) model and associated stimuli. (**A**) The set of 51 simple closed shapes from *Pasupathy and Connor, 2001*. Shapes are shown to relative scale. Shape size, given in pixels in the text, refers to the diameter of the big circle (top row, 2nd shape from the left). Each shape was shown at up to eight rotations as dictated by rotational symmetry, e.g., the small and large circles (upper left) were only shown at one rotation. This yielded a set of 362 unique shape stimuli. Stimuli were presented as white-on-black to the network (not as shown here). (**B**) Example shape with points along the boundary (red circles) indicating where angular position and curvature values were included in the APC model. (**C**) Points from the example shape in (**B**) are plotted in the APC plane where x-axis is angular position and y-axis is normalized curvature. Note the red circle furthest to the left at 0° angular position and negative curvature corresponds to the concavity at 0° on the example shape in (**B**). A schematic APC model is shown (ellipse near center of diagram) that is a product of Gaussians along the two axes. This APC model would describe a neuron with a preference for mild concavities at 135°.

(e.g., RF center). For this V4 neuron, the RF profile, measured by the mean response across all stimuli at each position (*Figure 6B*, green; see Materials and methods), falls off faster than the position-correlation function, consistent with a high degree of translation invariance.

A similar analysis for the example CNN unit, Conv2-113, reveals a steep drop-off in its position-correlation function (*Figure 6E*, red) compared to its RF profile (green). In particular, when stimuli were shown 13 pixels to the left of center (black arrow) the aggregate firing rate (see Materials and methods) was 87% of maximum, but the correlation was near zero. The largely

uncorrelated selectivity at two points within the RF indicates low translation invariance. Thus, despite its high APC r-value (*Figure 5A*), its low translation invariance diminishes it as a good model for V4 boundary contour tuning. This behavior was typical in layer Conv2, as demonstrated by the position-correlation function averaged across all units in the layer (*Figure 6F*). Specifically, the correlation (red) falls off rapidly compared to the RF profile (green) even for small displacements of the stimulus set.

For deeper layers, RFs tend to widen and translation invariance increases. This is exemplified by unit 369 in the fourth convolutional layer (*Figure 6G*) and the Conv4 layer average (*Figure 6H*): on average the correlation (red) more closely follows the RF profile (green) and does not drop to zero near the middle of the RF. In the deepest layers, exemplified by the FC7 unit from *Figure 5C*, the RFs become very broad (*Figure 6I*, green) and there is very little fall-off in correlation (red) even for shifts larger than the stimulus size. This is true for the layer average as well (*Figure 6J*). These plots show that shape selectivity becomes more translation invariant relative to RF size, and not just in terms of absolute distance, as signals progress to deeper layers.

To quantify translation invariance for each unit with a single number, we defined a metric, TI, based on the normalized average covariance of the response matrix across positions (see Materials and methods). The values of this metric, which would be one for perfect (and zero for no) correlation across positions, are shown for the example CNN units in *Figure 6E,G and I*. The trend for increasing TI with layer depth seen in *Figure 6* (panels F, H and J) is borne out in the cumulative distributions of TI broken down by CNN layer (*Figure 7A*). For comparison, the cumulative distribution of our TI metric for 39 V4 neurons from the study of *El-Shamayleh and Pasupathy (2016)* is plotted (red). Only the deepest four layers (Conv5 to FC8) had median TI values that approximated or exceeded that of our V4 population. Conv1 is excluded because its RFs are far too small to fully contain our stimuli at multiple positions (see Materials and methods). The substantial increase in TI for deeper layers is striking relative to the modest progression in APC r-values observed in *Figure 5G*.

An intuitive motivation for CNN architecture, chiefly convolution (repetition of linear filtering at translated positions) and max pooling, is the desire to achieve a translation invariant representation (*Fukushima, 1980*; *Rumelhart et al., 1986*; *Riesenhuber and Poggio, 1999*; *Serre et al., 2005*; *Cadieu et al., 2007*). This might lead to the idea that responses of units within these nets are translation invariant by design, but the observation that strong translation invariance only arises in later layers begins to deflate this notion. Furthermore, we computed TI for the same units and stimuli but in the untrained network. We found that the degradation of TI in an untrained network (*Figure 7B*) was even more dramatic than the degradation of APC tuning (*Figure 5H*). Specifically, it was very rare for any FC-layer unit in the untrained network to exceed the median TI values for those layers in the trained network.

To assess the influence of neuronal noise on our comparison of TI between V4 and AlexNet, we estimated an upper bound on how much TI could have been reduced by V4 response variability (see Materials and methods). TI tended to be less influenced by noise for neurons having higher TI, in particular the upward correction of the r-value was negatively correlated with the raw TI value ($r = -0.6$, $p < 0.001$, $n = 39$). Thus, for cells at the upper range of TI, we do not expect sampling variability to strongly influence our measurements. The distribution of V4 TI values corrected for noise is superimposed in *Figure 7A and B* (pink line). The modest rightward shift in the corrected distribution relative to the original raw distribution (red line) does not change our conclusion that only the deepest several layers in AlexNet have average TI values that match or exceed that of V4.

Our TI metric above was measured for horizontal stimulus shifts; however, we also measured TI for vertical shifts and verified that there was a high correlation between these two (r = 0.79) (*Figure 7—figure supplement 1*), particularly for high TI values.

## Identifying and visualizing preferences of candidate APC-like units

We now plot the joint distribution of our metrics for boundary contour tuning and translation invariance described above to identify candidate APC-like CNN units. *Figure 8* shows a unit square with APC r-value on the vertical axis and translation invariance, TI, on the horizontal axis. An ideal unit would be represented by the upper right corner, (1,1). The hypothetical best V4 neurons lie within this space at the red X (TI $= 0.97$, $r = 0.80$). This best V4 point is a hybrid of the observed highest APC r-value from the *Pasupathy and Connor (2001)* study, and the highest TI value from our re-

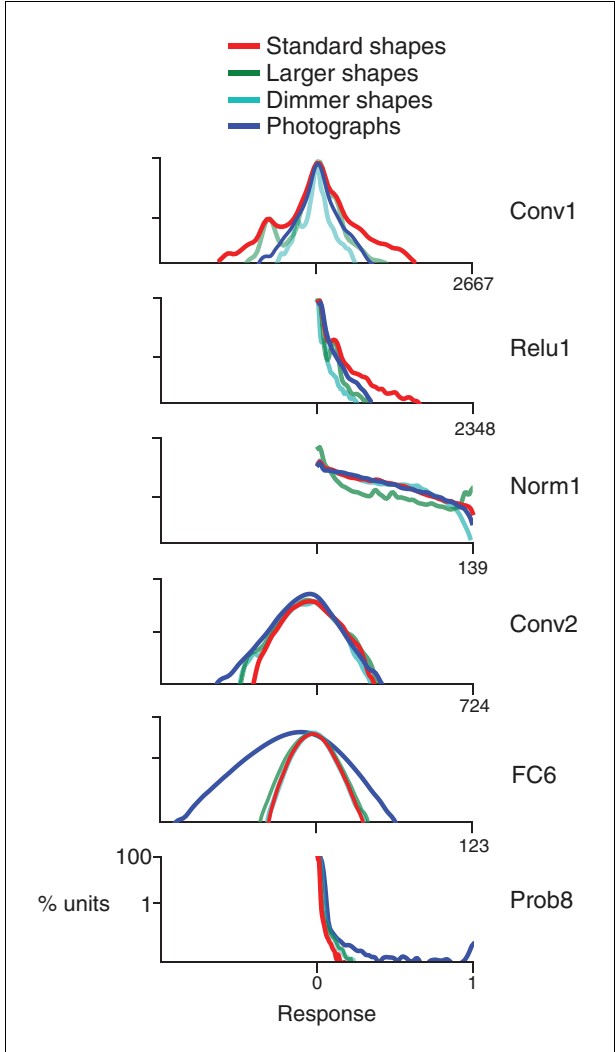

**Figure 4.** Response distributions for shapes and natural images in representative CNN layers. In each panel, the frequency distribution of the response values across all unique units in a designated CNN sublayer is plotted for four stimulus sets: our standard shape set (red; size 32 pixels, stimulus intensity 255, see Materials and methods), larger shapes (cyan; size 64 pixels, intensity 255), dimmer shapes (green; intensity 100, size 32 pixels) and natural images (dark blue). Natural images (n = 362, to match the number of shape stimuli) were pulled randomly from the ImageNet 2012 competition validation set. From top to bottom, panels show results for selected sublayers: Conv1, Relu1, Norm1, Conv2, FC6 and Prob8 (*Figure 2C* lists sublayer names). The number of points in each distribution is given by the number of stimuli (362) times the number of unique units in the layer (*Figure 2D*). The vertical axis is log scaled as most distributions have a very high peak at 0. For Conv1, standard shapes drove a wider overall dynamic range than did images because of the high intensity edges that aligned with parts of the linear kernels (*Figure 1*). This was not the case for larger shapes because they often over-filled the small Conv1 kernels. For Relu1, negative responses are removed by rectification after a bias is added. At Conv2, there is little difference between the four stimulus sets on the positive side of the distribution. This changes from FC6 forward, where natural images drive a wider range of responses. For Prob8, natural images (dark blue line) sometimes result in high probabilities among the 1000 categorical units, whereas shapes do not.

The online version of this article includes the following figure supplement(s) for figure 4:

**Figure supplement 1.** Sparsity of CNN and V4 unit responses to shape stimuli (see end of document).

analysis of the *El-Shamayleh and Pasupathy (2016)* data. In comparison, the most promising CNN unit lies at the orange star (TI = 0.91, *r* = 0.77), very close to the hypothetical best V4 point. To demonstrate how the CNN population falls on this map, we plotted 100 randomly chosen units from an early layer, Conv2 (dark brown), and a deep layer, FC7 (orange). Although only a few FC7 units

approach the hypothetical best V4 point, many units are better than the average V4 neuron (red lines, *Figure 8*). In contrast, most units from Conv2 are much further from ideal V4 behavior, but they span a large range, indicating that even in the second convolutional layer, some units have ended up, after training, having high TI and high APC r-values.

To determine whether units identified as being the most APC-like, that is those closest to (1,1) in *Figure 8*, respond to natural images in a manner qualitatively consistent with boundary curvature selectivity in an objected-centered coordinate frame, we identified image patches that were most facilitatory (drove the greatest positive responses) and most suppressive (drove the greatest negative responses) for the 50,000 image test-set from the 2012 ImageNet competition. We then used a visualization technique (*Zeiler and Fergus, 2013*) to project back ('deconvolve') from the unit onto each input image through the connections that most strongly contributed to the response, thereby revealing the regions and features supporting the response. We examined the ten most APC-like units in each of seven layers from Conv2 to FC8. Below we describe major qualitative observations as a function of layer depth.

Visualizing the ten most APC-like units in Conv2 revealed selectivity for orientation, conjunctions thereof, or other textures. For example, unit Conv2-113 (from *Figures 5A* and *8E*), was best driven by lines at a particular orientation (*Figure 9A*) and most suppressed by oriented texture running roughly orthogonal to the preferred. This explains why this unit responded well only to shapes that have long contours extending to a point at the upper left, and poorly to shapes having a broad convexity or concavity to the upper left (*Figure 5B*). Another Conv2 example (*Figure 9B*) was driven best by the conjunction of a vertical that bends to the upper left and a horizontal near the top of the RF that meet at a point in the upper left. Examining the input images reveals that textures and lines (e.g., the bedspread and rocking chair cushion) are as good at driving the unit as are boundaries of objects. A third unit (*Figure 9C*) preferred conjunctions of orientations and was suppressed by lines running orthogonal to the preferred vertical orientation. The preferred pattern was usually not an object boundary, but could surround negative space or be surface texture. These observations, taken together with the poor translation invariance of Conv2 relative to deeper layers, suggest that units at this early stage are not coding boundary conformation in an object-centered way, but that any pattern matching the preferred features of the unit, regardless of its position with respect to an object, will drive these units well.

From Conv3 to Conv5, the visualizations of the most APC-like units were more often consistent with an encoding of portions of object boundaries. Unit Conv3-156 was driven best by the broad downward border of light objects (*Figure 10A*), particularly dog paws. The most suppressive features for this 'downward-dog-paw' unit were dark regions, often negative space, with relatively straight edges. The deconvolved features tended to emphasize the lower portion of the object border. A similar example, Conv3-020, had a preference for the upper border of bright forms (e.g., flames; *Figure 10B*) and was suppressed by the upper border of dark forms (often dark hair on heads). This unit was representative of a tendency for selectivity for bright regions with broad convexities (e.g., Conv4-171, not shown). We assume that more dark-preferring units would have been found had our stimuli been presented as black-on-white. These trends continued with greater category specificity in Conv5. For example, Conv5-161 was driven best by the rounded, convex tops of white dog heads (*Figure 10C*), including some contribution from the eyes, and was most suppressed by human faces below the eyebrows. Unit Conv5-144 was best driven by the upward facing points of the tops of objects, particularly wolf ears and steeples (*Figure 10D*). This 'wolf-ear-steeple' unit was most suppressed by rounded forms, and may be important for distinguishing between the many dog categories with and without pointed ears. In addition to units like these, which appeared to be selective for portions of boundaries, there were several units that appeared to detect entire circles (*Figure 11*), and thus fit well to an APC model with specificity for curvature but broadly accepting of any angular position.

In the FC layers, the most excitatory images were revealing about unit preferences, but the deconvolved features provided less insight because power in the back projection was typically widely distributed across the input image. For example, unit FC6-3030 (*Figure 12A*) responded best to hourglasses, but deconvolution did not highlight a particular critical feature. The shape stimuli driving the highest and lowest five responses (*Figure 12A*, bottom row) suggest that a cusp (convexity) facing upward is a critical feature, consistent with the APC model fit (*Table 1*). The most suppressive natural images (not shown) were more diverse than those for the Conv layers, and thus provided

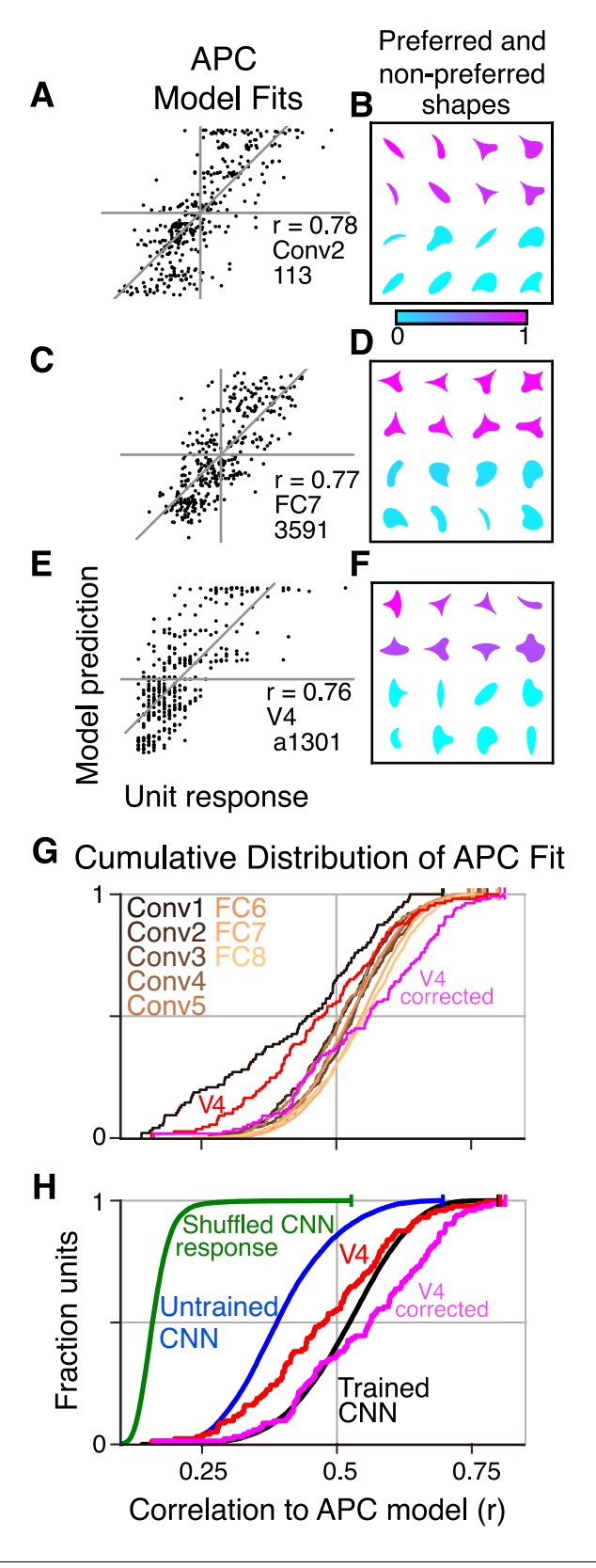

**Figure 5.** Boundary curvature selectivity for CNN units compared to V4 neurons. (**A**) APC model prediction vs. CNN unit response for an example CNN unit from an early layer (Conv2-113). (**B**) The top and bottom eight

*Figure 5 continued on next page*

*Figure 5 continued*

shapes sorted by response amplitude (most preferred shape is at upper left, least at lower right) reveal a preference for convexity to the upper left (such a feature is absent in the non-preferred shapes). This is consistent with the APC fit parameters, $\mu_c = 1.0$, $\sigma_c = 0.53$, $\mu_a = 135°$, $\sigma_a = 23°$. (C) Predicted vs. measured responses for another well-fit example CNN unit (FC7-3591) but in a later layer. (D) Top and bottom eight shapes for example unit in (C). The APC model fit was $\mu_c = -0.1$, $\sigma_c = 0.15$, $\mu_a = 112°$, $\sigma_a = 44°$. (E) Model prediction vs. neuronal mean firing rate (normalized) for the V4 neuron (a1301) that had the highest APC fit r-value. (F) The top eight shapes (purple) all have a strong convexity to the left, whereas the bottom eight (cyan) do not. The APC model fit was $\mu_c = 1.0$, $\sigma_c = 0.39$, $\mu_a = 180°$, $\sigma_a = 23°$. (G) The cumulative distributions (across units) of APC r-values are plotted for the first sublayer of each major CNN layer (boldface names in *Figure 2C*) from Conv1 (black) to FC8 (lightest orange). The other sublayers (distributions not shown for clarity) tended to have lower APC r-values but the trend for increasing APC r-value with layer was similar. For comparison, red line shows cumulative distribution for 109 V4 neurons (*Pasupathy and Connor, 2001*), and pink line shows V4 distribution corrected for noise (see Materials and methods). (H) The cumulative distribution of r-values for the APC fits for all CNN units (black), CNN units with shuffled responses (green), units in an untrained CNN (blue) and V4 (red and pink). The far leftward shift of the green line shows that fit quality deteriorates substantially when the responses are shuffled across the 362 stimuli within each unit.

little direct insight. Broadly, many of the top ten APC-like units in the FC layers fell into two categories: those preferring images with rounded borders facing approximately upwards (we refer to these as the 'balls' group) and those associated with a concavity between sharp convexities, also facing approximately upwards (the 'wolf-ears' group). For example, FC7-3192 (*Figure 12B*) responded best to images of round objects (e.g., golf balls) and to shapes having rounded tops. FC7-3591 (*Figure 12C*), which was the most APC-like unit by our joint TI-APC index (orange star in *Figure 8*), responded best to starfish and rabbit-like ears pointing up. Shapes with a convexity at 112° drove the unit most strongly, whereas shapes with rounded tops and overall vertical orientation yielded the most negative responses. FC7-3639 (*Figure 12D*) is an example of a wolf-ears unit, and its preferred shapes include those with a convexity pointing upwards flanked by one or two sharp points. In FC8, where there is a one-to-one mapping from units onto the 1000 trained categories, the top ten APC units were evenly split between the wolf-ears group (categories: kit fox, gray fox, impala, red wolf and red fox) and the balls group (categories: ping-pong balls, golf balls, bathing caps, car mirrors and rugby balls). For example, unit FC8-271 (*Figure 12E*) corresponds to the red wolf category and units FC8-433 and FC8-722 correspond to the bathing cap and ping-pong ball categories, respectively.

What is most striking about the deep-layer (FC) units is that, in spite of their tendency to be more categorical, that is to respond to a wolf in many poses or a ping-pong ball in many contexts, they still showed systematic selectivity to our simple shapes. We hypothesized that these FC units were driven by a range of image properties that correlated well with the target category, and that shape was simply one among others such as texture and color. We examined how much better the units were driven by the best natural images compared to our best standard shapes. *Figure 13* shows for the top-10 APC-like units in each layer, that the best image drove responses on average about 2 times higher than did the best shape for Conv2-4, about 4–5 times higher for FC6-7 and more than 8 times higher for FC8. This is consistent with the hypothesis that shape tuned mechanisms contribute to the selectivity of these units, but are not sufficient in the absence of other image properties to drive the FC layers strongly. Nevertheless, the selectivity for simple shapes at the final layer appears to be qualitatively consistent with the category label. Notably, only two APC-like units responded better to a shape than to any natural image, but both were Conv4 units selective for bright circular regions (not shown), and the best stimulus was our large circle (*Figure 3A*, second from upper left).

## CNN fit to V4 responses

Above, we examined the ability of CNN units to approximate the boundary curvature selectivity of V4 neurons as described by the APC model, but while an APC model provides a good description of the responses of many V4 neurons, there are also neurons for which it explains little response variance across our shape set. We therefore examined whether the CNN units might directly provide a better fit (than the APC model) to the responses of the V4 neurons. We used cross-validation (see

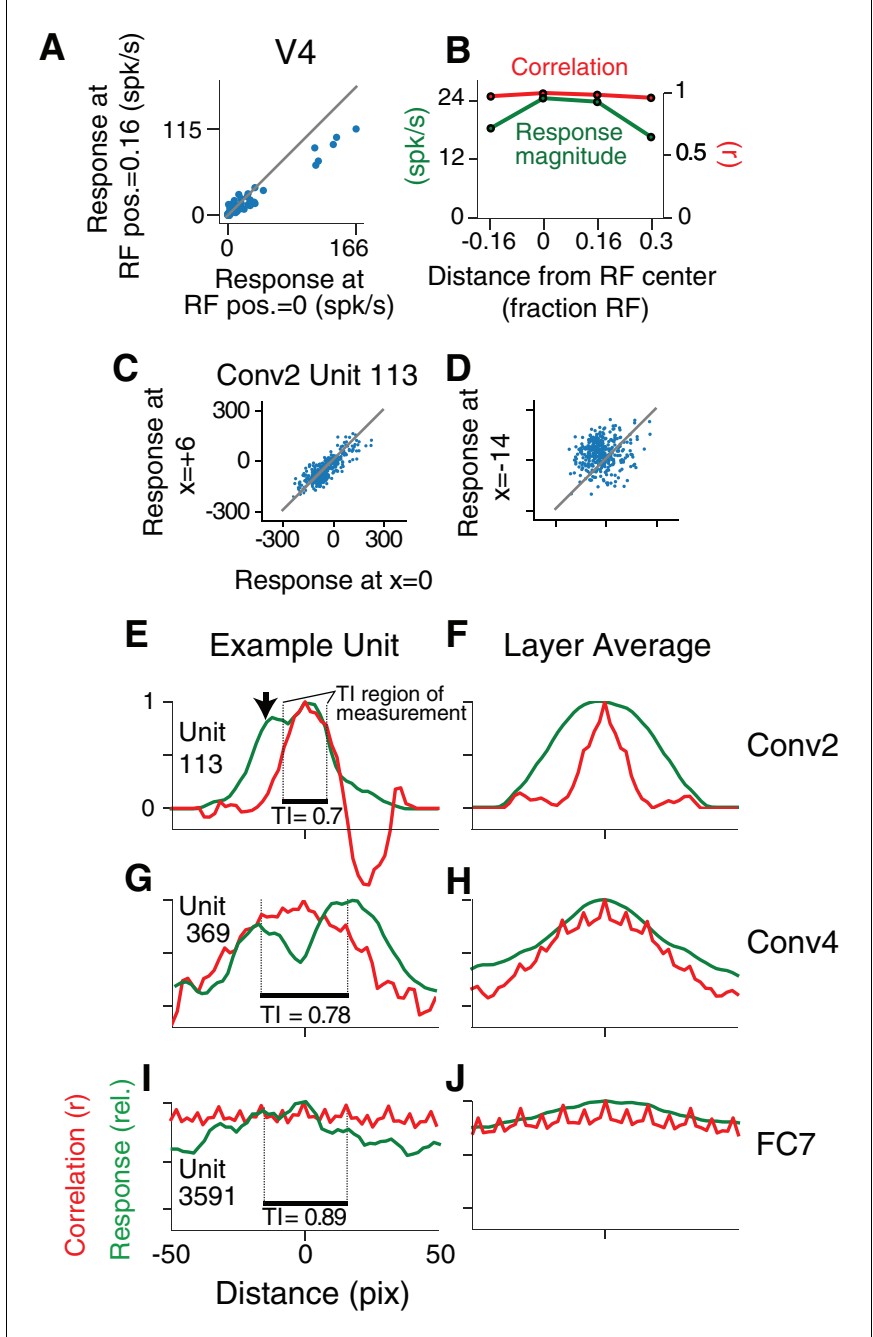

**Figure 6.** Translation invariance as a function of position across the RF. (**A**) For an example neuron from the V4 study of **El-Shamayleh and Pasupathy (2016)**, the responses to stimuli shifted away from the RF center by 1/6 of the estimated RF size are plotted against those placed in the RF center. The overall response magnitude decreases with shift, but a strong linear relationship is maintained between responses at the two positions. (**B**) In green, the RF profile of the same neuron from (**A**) is plotted (average response at each position). In red, the correlation of the responses at each position with the responses at RF center. (**C**) For unit Conv2-113, responses to stimuli shifted six pixels to the right are plotted against responses for centered stimuli. (**D**) For the same unit in (**C**), responses for stimuli shifted 14 pixels to the left vs. responses for centered stimuli. (**E**) For unit Conv2-113, the position-correlation function is plotted in red. The RF profile, that is the normalized response magnitude (square root of sum of squared responses) across all shapes is plotted in green. The region over which TI is measured, where all stimuli are wholly within the CRF (see Materials and methods), is within dotted lines bookending horizontal black bar. The unit is less translation invariant because it continues to have a large response even when correlation drops quickly from center. This is reflected in the lower TI score of 0.7. (**F**) The averages of the

*Figure 6 continued on next page*

*Figure 6 continued*

correlation and RF profiles across all units in the Conv2 layer show that correlation drops off much more rapidly than the RF profile. (**G**) Same as in (**E**) but for a unit in the 4th convolutional layer (Conv4-369). There is a broadened correlation profile compared to the Conv2 unit. (**H**) For Conv4, the average position-correlation function (red) has a wider peak than that for Conv2, more closely matching the shape of the average RF profile (green). It also has serrations that occur eight pixels apart, which corresponds to the pixel stride (discretization) of Conv2 (*Figure 2A*; see Materials and methods). (**I**) The shape-tuned example unit FC7 3591 (*Figure 5C*) in the final layer is highly translation invariant (TI = 0.89). (**J**) The response profile and correlation stay high across the center of the input field on average across units in FC7.

Materials and methods) to put these very different models on equal footing. *Figure 14* shows the cross-validated, best fit r-values for the APC model plotted against those for the CNN units. Neither model is clearly better on average: just over half (56/109) of neurons were better fit by the APC model, while just under half (53/109) were better fit by a CNN unit. Only 21 of 109 neurons had significant deviations from the line of equality (*Figure 14*, red) and these were evenly split: 11 better fit by the APC model and 10 by the CNN. The similar performance of the APC model and CNN could be a result of the CNN and APC model explaining the same component of variance in the data, or explaining largely separate components of the variance. To assess this, for each V4 neuron, we removed from its response the component of variance explained by its best-fit APC model. For this

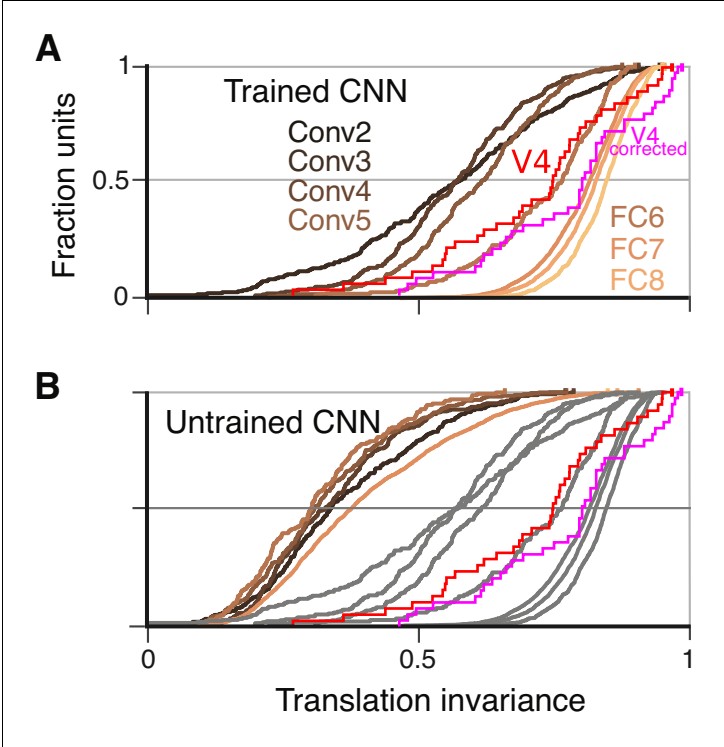

**Figure 7.** Cumulative distributions of the TI metric for the CNN and V4. (**A**) The cumulative distributions (across units) of TI are plotted for the first sublayer of each major CNN layer (boldface names in *Figure 2C*) from Conv2 (black) to FC8 (lightest orange). There is a clear increase in TI moving up the hierarchy. The TI distribution for V4 is plotted in red, and an upper bound for noise correction is plotted in pink (see Materials and methods). The other sublayers (distributions not shown for clarity) tended to have lower TI values but the trend for increasing TI with layer was similar. (**B**) The cumulative distribution of TI across layers in the untrained CNN. There is a large shift toward lower TI values in comparison to the trained CNN (faint grey and red and pink lines reproduce traces from panel A).

The online version of this article includes the following figure supplement(s) for figure 7:

**Figure supplement 1.** The consistency of translation invariance across sampling directions.

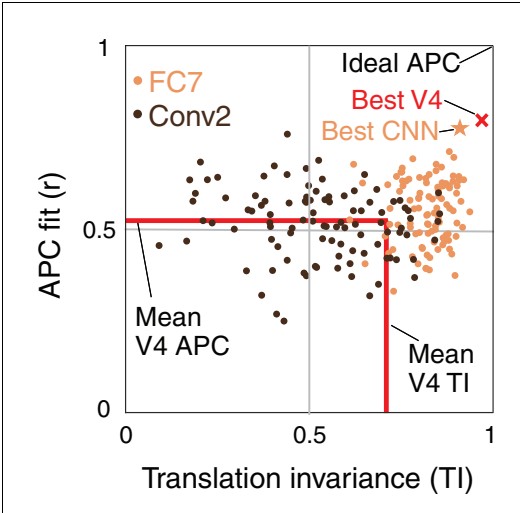

**Figure 8.** Summary of the similarity of CNN units to V4 neurons in terms of translation invariance (TI) and fit to the APC model. For 100 randomly selected CNN units from Conv2 (brown) and FC7 (orange), APC r-value is plotted against TI. The hypothetical highest scoring V4 unit (red ×) is the combination of the highest TI score and the highest APC fit from separate V4 data sets (0.97, 0.80). The highest scoring unit in the CNN (FC7-3591, from *Figure 5C*, *Figure 6I* and *Figure 12C*) is indicated by the orange star (0.91, 0.77) and is close to the hypothetical best V4 unit. The red lines indicate the mean V4 values along each axis, not including any correction for noise (see *Figures 5* and *7* for estimated noise correction, pink lines).

APC-orthogonalized V4 response, the CNN model had a median correlation to V4 of $r = 0.29$ (SD = 0.11), much lower than the APC model's $r = 0.47$ (SD = 0.12) median . For 94/109 neurons, the APC model explained more variance than the variance uniquely explained by the CNN. Overall, we conclude that the APC model and the CNN explain similar features of V4 responses for most neurons.

## Discussion

We examined whether the CNN known as Alex-Net, designed to perform well on image classification, contains units that appear to have boundary curvature selectivity like that of V4 neurons in the macaque brain. Although our simple shape stimuli were never presented to the network during training, we found that many units in the CNN were V4-like in terms of quantitative criteria for translation invariance and goodness of fit to a boundary curvature model. While units throughout AlexNet had good fits to the APC model, relatively poor translation invariance in the early layers meant that only the middle to deeper layers had substantial numbers of units that came close to matching exemplary APC-tuned V4 neurons. Based on our quantitative criteria and on the qualitative visualization of preferred features identified in natural images, we believe that APC-like units within middle layers of trained CNNs currently provide the best image-computable models for V4 boundary curvature selectivity.

Finding such matches at the single unit level is striking because the deep net and our macaques differ dramatically in their inputs, training and architecture. The animals never saw ImageNet images and probably never saw even a single instance of the overwhelming majority of the 1000 output categories of AlexNet. They did not see the forest, ocean, sky nor other important contexts for AlexNet categories, nor had AlexNet been trained on the artificial shapes used to characterize V4. While the macaque visual system may be shaped by real-time physical contact with a 3D dynamic world, Alex-Net cannot and was not even given information about the locations nor boundaries of the targets to be classified within its images during categorization training. AlexNet lacks a retina with a fovea, an LGN, feedback from higher areas, dedicated excitatory and inhibitory neurons, etc., and it does not have to compute with action potentials. Our results suggest that image statistics related to object boundaries may generalize across a wide variety of inputs and may support a broad variety of tasks, thereby explaining the emergence of similar selectivity in such disparate systems.

### Visualization of V4-like CNN units

By applying a CNN visualization technique to APC-like units identified by our quantitative criteria, we found that some of these CNN units appeared, qualitatively, to respond to shape boundaries in natural images whereas many others did not. In early layers, particularly Conv2, the strongest responses were not driven specifically by object boundaries but instead by other image features including texture, accidental contours and negative space. In contrast, candidate APC units in intermediate layers often responded specifically to natural images patches containing object boundaries, suggesting that these units are APC-like. In the deeper (FC) layers, units were poorly driven by our shape stimuli relative to natural images, and the preferred natural images for a given unit appeared

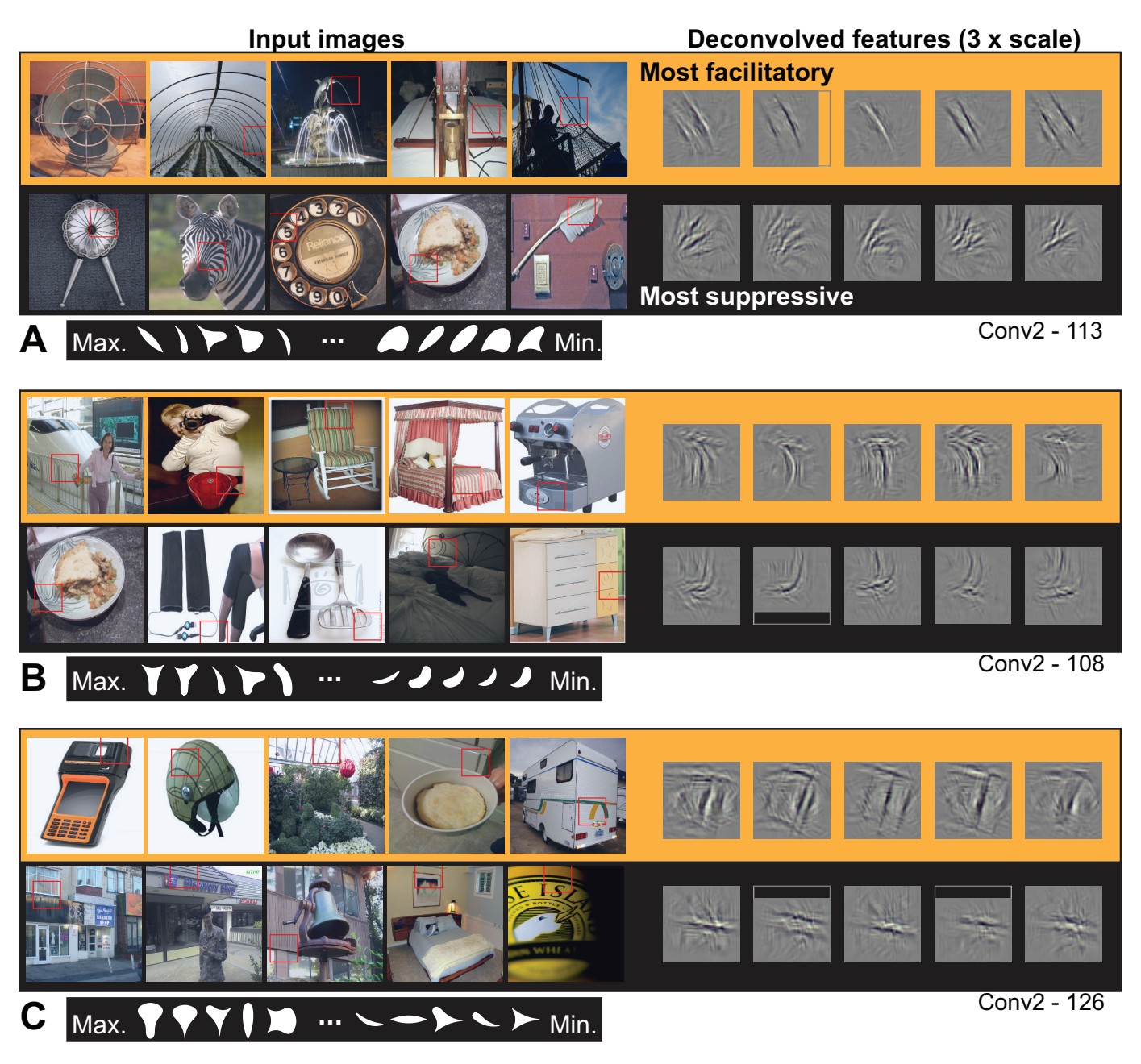

**Figure 9.** Visualization of APC-like units in layer Conv2. (**A**) For unit Conv2-113, the five most excitatory image patches are indicated by red squares superimposed in the raw images (top row, left side, from left to right). The size of the red square corresponds to the maximal extent of the image available to Conv2 units (see *Figure 2B*). In corresponding order, the five deconvolved features are shown at the upper right, with a 3x scale increase for clarity. The blank rectangular region at the right side of the second feature indicates that this part of the unit RF extended beyond the input image (such regions are padded with zero during response computation). For the same unit, the lower row shows the five most suppressive image patches and their corresponding deconvolved features. We examined the top 10 most excitatory and suppressive images, and for all examples in this and subsequent figures, they were consistent with the top 5. Below the natural images are the top 5 and bottom five shapes (white on black background) in order of response from highest (at left) to lowest (at right). Shapes are shown at 2x scale relative to images, for visibility. (**B**) Same format as (**A**), but for unit Conv2-108. (**C**) Same format as (**A**), but for unit Conv2-126. In all examples, the most suppressive features (bottom row in each panel) tend to run orthogonal to, and at the same RF position, as the preferred features (top row in each panel) For APC fit parameters, see *Table 1* in Results text. Thumbnails featuring people were redacted for display in the published article, in line with journal policy. Input image thumbnails were accessed via the ImageNet database and the original image URLs can be found through this site: http://image-net.org/about-overview.

*Figure 9 continued on next page*

*Figure 9 continued*

similar along many feature dimensions (e.g., texture, background context) beyond simply the curvature of object boundaries. We speculate that these units are jointly tuned to many features and that object boundaries alone account for only part of their tuning. More work is needed to understand the FC-layer units with high APC r-values; however, we believe units in the middle layers, Conv3-5, provide good working models for understanding how APC-tuning might arise from earlier representations, how it may depend on image statistics and how it could support downstream representation.

## Training and translation invariance

Training dramatically increased the number of units with V4-like translation invariance, particularly in the FC layers (*Figure 7A* vs. *Figure 7B*). Since the trained and untrained nets have the same architecture, the increase in TI is not simply a result of architectural features meant to facilitate translation invariance, for example max-pooling over identical, shifted filters. Thus, while CNN architecture is often associated with translation invariance (*Fukushima, 1980*; *Rumelhart et al., 1986*; *Riesenhuber and Poggio, 1999*; *Serre et al., 2005*; *Cadieu et al., 2007*; *Goodfellow et al., 2009*; *Lenc and Vedaldi, 2014*), we find that high TI for actual single unit responses is only achieved in tandem with the correct weights. We are currently undertaking an in-depth study comparing the trained and untrained networks to elucidate statistical properties of weight patterns that support translation invariance. Our preliminary analyses show that spatial homogeneity of a unit's kernel weights across features correlates with its TI score, but this correlation is weaker in higher layers. Alternative models of translation-invariant tuning in V4 include the spectral receptive field (SRF) model (*David et al., 2006*) and HMax model (*Cadieu et al., 2007*). The former made use of the Fourier spectral power, which is invariant to translation of the input image, but this phase insensitivity prevents the SRF model from explaining APC-like shape tuning (*Oleskiw et al., 2014*). The HMax model of

**Table 1.** Fit parameters and TI metric for example CNN units.
Unit numbers are given starting at zero in each sublayer. The APC model parameters, $\mu_c$, $\sigma_c$, $\mu_a$ and $\sigma_a$, correspond to those in *Equation 2*. The TI metric is given by *Equation 3*. For visualization of preferred stimuli for example units, see *Figures 9–12*.

| Layer | Unit | APC r | $\mu_c$ | $\sigma_c$ | $\mu_a$ | $\sigma_a$ | TI |
|-------|------|-------|---------|------------|---------|------------|-----|
| Conv2 | 108 | 0.67 | 0.7 | 0.72 | 134 | 34 | 0.76 |
| Conv2 | 113 | 0.76 | 0.9 | 0.39 | 134 | 22 | 0.70 |
| Conv2 | 126 | 0.67 | 0.1 | 0.72 | 337 | 51 | 0.81 |
| Conv3 | 20 | 0.68 | 0.5 | 0.01 | 224 | 171 | 0.90 |
| Conv3 | 156 | 0.67 | 0.5 | 0.01 | 337 | 171 | 0.79 |
| Conv3 | 334 | 0.73 | 0.2 | 0.12 | 157 | 171 | 0.74 |
| Conv4 | 203 | 0.71 | 0.2 | 0.16 | 292 | 171 | 0.77 |
| Conv5 | 144 | 0.65 | 0.9 | 0.29 | 89 | 30 | 0.89 |
| Conv5 | 161 | 0.72 | 0.2 | 0.16 | 112 | 87 | 0.85 |
| FC6 | 3030 | 0.73 | −0.1 | 0.16 | 89 | 26 | 0.89 |
| FC7 | 3192 | 0.75 | 0.2 | 0.16 | 112 | 171 | 0.91 |
| FC7 | 3591 | 0.78 | −0.1 | 0.16 | 112 | 44 | 0.89 |
| FC7 | 3639 | 0.76 | −0.1 | 0.16 | 112 | 114 | 0.92 |
| FC8 | 271 | 0.73 | −0.1 | 0.16 | 112 | 114 | 0.91 |
| FC8 | 433 | 0.70 | 0.3 | 0.21 | 112 | 130 | 0.91 |
| FC8 | 722 | 0.72 | 0.2 | 0.08 | 112 | 130 | 0.93 |

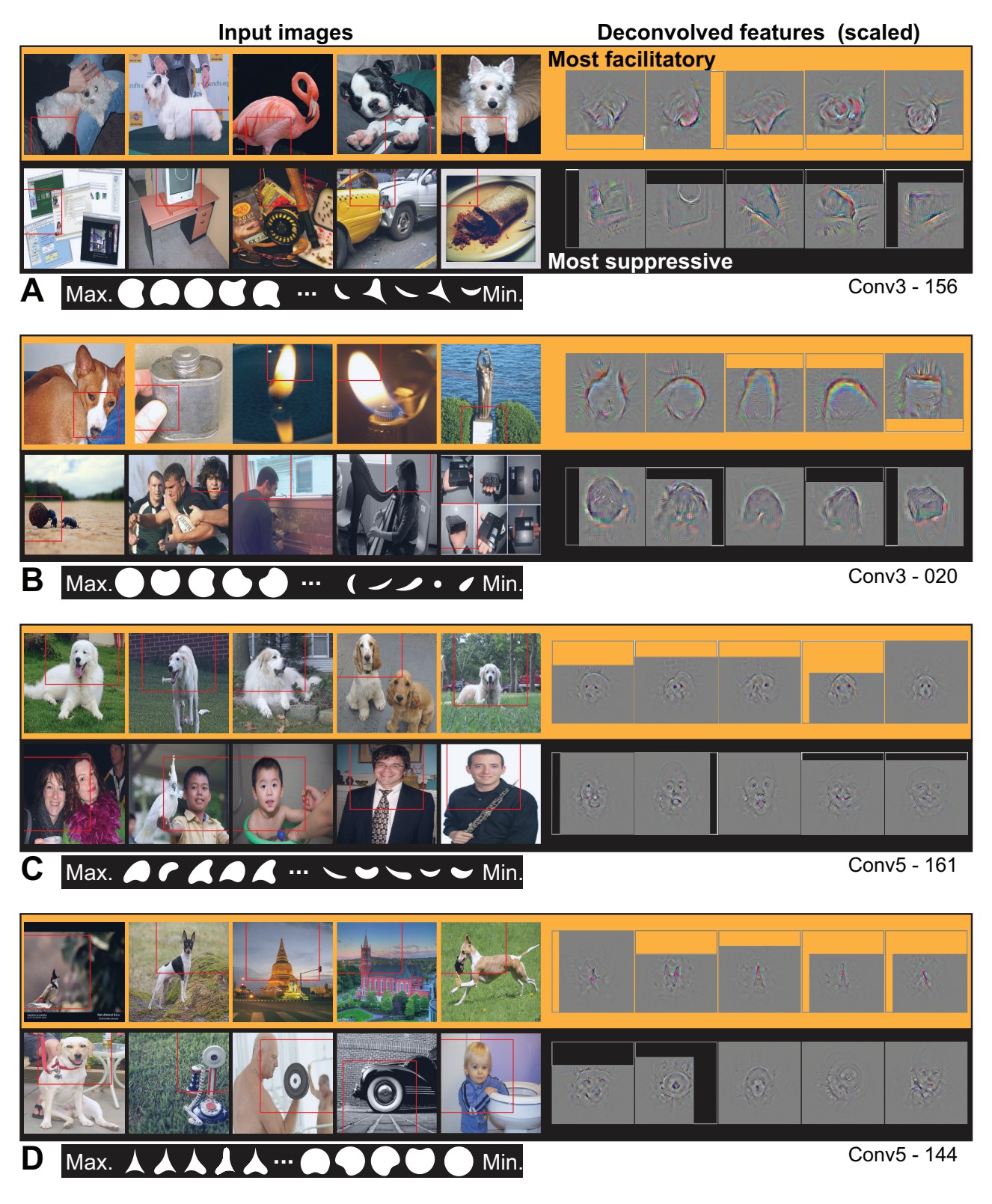

**Figure 10.** Visualization of APC-like units in layers Conv3 to Conv5. (A) Visualization for unit Conv3-156, using the same format as *Figure 9*. Deconvolved features are scaled by 1.8 for visibility. (B) Same as (A), for unit Conv3-020. (C) Same for unit Conv5-161, but deconvolved features are scaled by 1.15. (D) Same as (C), but for unit Conv5-144. For APC fit parameters, see *Table 1* in main text. Thumbnails featuring people were redacted for display in the published article, in line with journal policy. Input images were accessed via the ImageNet database and the original image URLs can

*Figure 10 continued on next page*

*Figure 10 continued*

be found through this site: http://image-net.org/about-overview.

*Cadieu et al. (2007)* is a shallower network with the equivalent of two convolutional layers and does not achieve the strong translation invariance found in deeper layers here (*Popovkina et al., 2017*). Overall, translation invariance at the single-unit level is not a trivial result of gross CNN architecture, yet it is crucial for modeling V4 form selectivity.

## Other studies of TI in CNNs

Although other studies have examined translation invariance and related properties (rotation and reflection invariance) in artificial networks (*Ranzato et al., 2007*; *Goodfellow et al., 2009*; *Lenc and Vedaldi, 2014*; *Zeiler and Fergus, 2013*; *Fawzi and Frossard, 2015*; *Güçlü and van Gerven, 2015*, *Shang et al., 2016*; *Shen et al., 2016*; *Tsai and Cox, 2015*), we are unaware of any study that has quantitatively documented a steady layer-to-layer increase of translation invariant form selectivity, measured for single units, across layers throughout a network like AlexNet. For example, using the invariance metric of *Goodfellow et al. (2009)*, *Shang et al. (2016)* (their Figure 4c) averaged over multiple types of invariance (e.g., translation, rotation) and over all units within a layer and found a weak, non-monotonic increase in invariance across layers in a CNN similar to AlexNet. Using the same metric but different stimuli, *Shen et al., 2016* found no increase and no systematic trend in invariance across layers of their implementation of AlexNet (their Figure 5). Although *Güçlü and van Gerven (2015)* plot an invariance metric against CNN layer, their metric is the half-width of a response profile, and thus it is unlike our TI selectivity metric. In spite of the importance of translation invariance in visual processing and deep learning (*LeCun et al., 2015*), there currently is no standard practice for quantifying it. An important direction for future work will be to establish standard and robust methods for assessing translation invariance and other transformation invariances to facilitate comparisons across artificial networks and the brain.

## Comparison to previous work

One way our approach to comparing the representation in a CNN to that in the brain differs from previous work is that we examined the representation of specific visual features at the single-unit level, whereas previous studies took a population level approach. For example, *Yamins et al. (2014)* modeled IT and V4 recordings using weighted sums over populations of CNN units, and *Khaligh-Razavi and Kriegeskorte, 2014* examined whether populations of CNN units represented categorical distinctions similar to those represented in IT (e.g., animate vs. inanimate). Also, *Kubilius et al. (2016)* examined whether forms perceived as similar by humans had similar CNN population representations. Our work is the first to quantitatively compare the single-unit representation in a CNN to that in a mid-level visual cortical area. We tested whether an artificial network matched the neural representation at a fundamental level—the output of single neurons, which are conveyed onward to hundreds or thousands of targets in multiple cortical areas. Unlike previous studies, we focused on specific physiological properties (boundary curvature tuning and translation invariance) with a goal of finding better models where a robust image-computable model is lacking. Furthermore, we use visualization of unit responses to natural images to qualitatively validate whether the representation that these response properties are intended to capture (an object-centered representation of boundary) does in fact hold across natural images. We believe this level of model validation, which includes quantitative and conceptual registration to documented neuronal selectivity, pushes the field beyond what has been done before. Our results allow modelers to focus on specific neural selectivities and work with concrete, identified circuits that have biologically plausible components.

Another major difference with prior work is that we fit the CNN to the APC model as opposed to directly to neural responses. This might seem like an unnecessary layer of abstraction, but the purpose of a model is not just predictive power but also interpretability, and the CNN's complexity runs counter to interpretability. The CNN is necessarily complex in order to encode complex features from raw pixel values, whereas the APC model has five interpretable parameters. The APC model

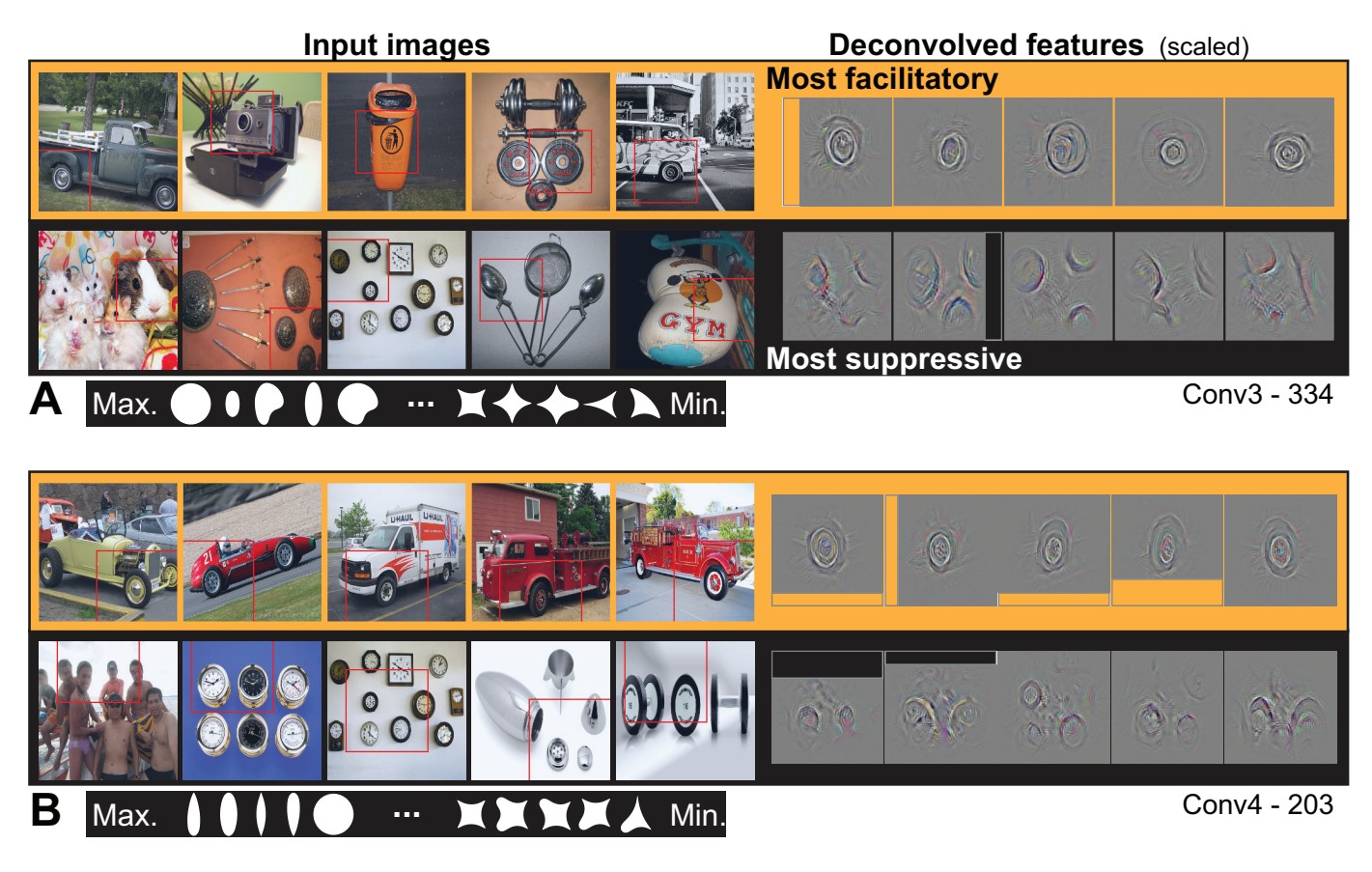

**Figure 11.** Visualization of APC-like units: circle detectors. These examples are representative of many units that were selective for circular forms. (**A**) Unit Conv3-334 was selective for a wide variety of circular objects near its RF center and was suppressed by circular boundaries entering its RF from the surround. Deconvolved feature patches are scaled up by 1.8 relative to raw images. (**B**) Unit Conv4-203 was also selective for circular shapes near the RF center, but showed category specificity for vehicle wheels. Suppression was not category specific but was, like that in (**A**), related to circular forms offset from the RF center. The higher degree of specificity in (**B**) is consistent with this unit being deeper than the example in (**A**). Deconvolved features are scaled by 1.4 relative to raw images. APC fit parameters are given in *Table 1*. Thumbnails featuring people were redacted for display in the published article, in line with journal policy. Input images were accessed via the ImageNet database and the original image URLs can be found through this site: http://image-net.org/about-overview.

describes responses to complex features while ignoring the details of how those features were computed from an image. By identifying APC-tuned units in the CNN, we gain an image-computable model of neural responses to interpretable features; these units can be studied to understand how and why such response patterns arise. When we separately tested whether the CNN units were able to directly fit the responses of V4 neurons, we found they were no better on average than the APC model, thus for a gain in interpretability, we did not suffer an overall loss of predictive power. Nevertheless, some V4 neurons were better fit directly to a CNN unit than to any APC model, suggesting there may be V4 representations beyond APC tuning that can be synergistically studied with CNNs.

## Value of artiphysiology

Comparing artificial networks to the brain can serve both computer and biological vision science (*Kriegeskorte, 2015*). What can an electrophysiologist learn from this study? First, our results demonstrate that there may already exist image-computable models for complex selectivity that match single-neuron data better than hand-designed models from neuroscientists. Second, finding matches

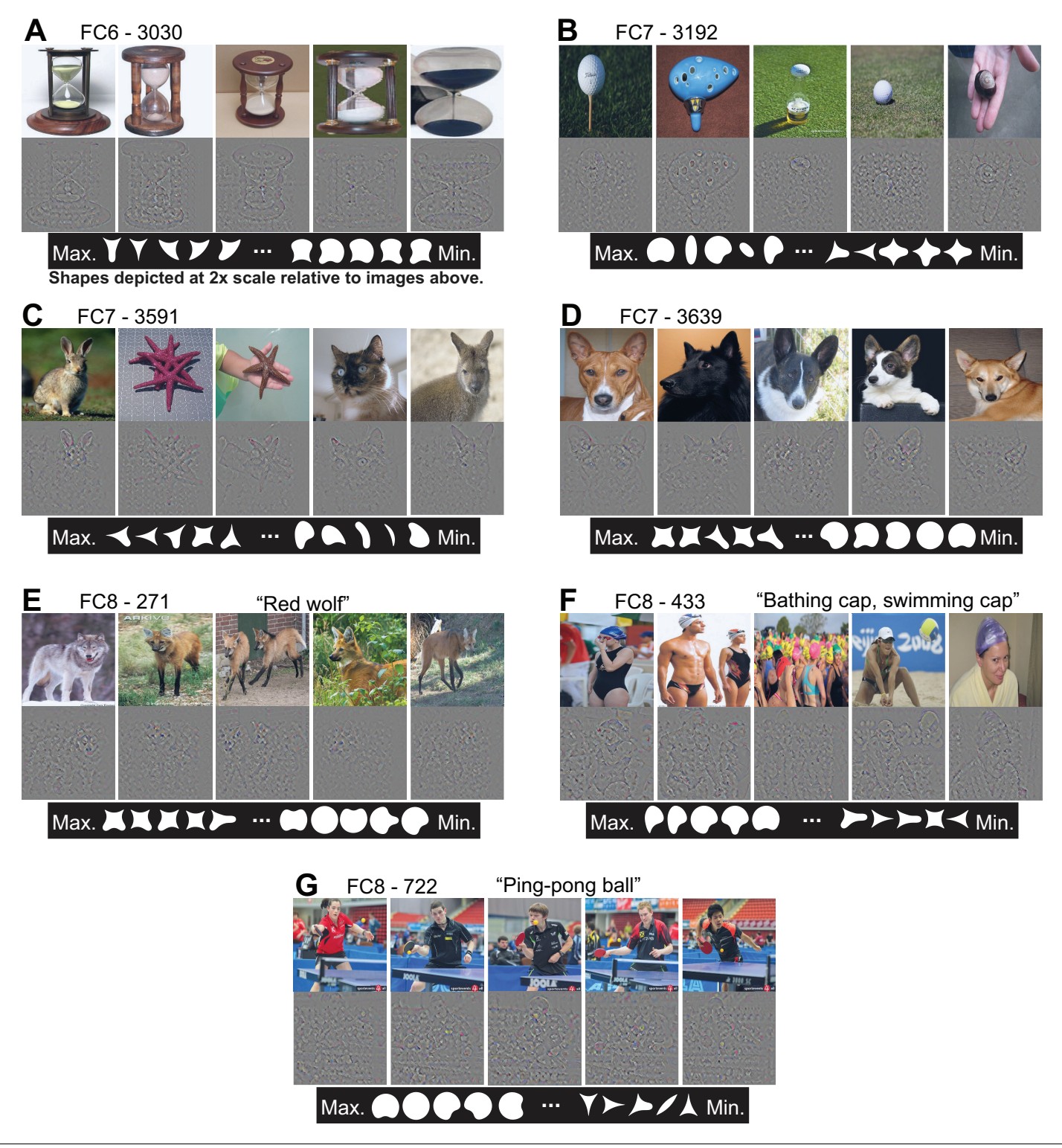

**Figure 12.** Visualization of APC-like units in the FC layers. (**A**) For unit FC6-3030, the top five images from the test set are shown above their deconvolved feature maps. The maximal RF for all FC units includes the entire image. At bottom, the top five shapes are shown in order from left to right, followed by the bottom five shapes such that the shape associated with the minimum response is the rightmost. For visibility, shapes are shown here at twice the scale relative to the images. (**B**) For unit FC7-3192, same format as (**A**). (**C**) For unit FC7-3591, same format as (**A**). (**D**) For unit FC7-3639, same format as (**A**). (**E**) For unit FC8-271, same format as (**A**), except the category of this output-layer unit is indicated as 'Red wolf.' (**F**) For unit FC8-433, same format as (**E**). (**G**) For unit FC8-722, same format as (**E**). See *Table 1* for APC fit values for all units. Thumbnails featuring people were

*Figure 12 continued on next page*

between neuronal selectivity in the brain and artificial networks trained on vast amounts of natural data provides one method for electrophysiologists to validate their findings. For example, our findings support the hypothesis that an encoding of boundary curvature in single units may be generally important for the representation of objects. Third, once a match is found based on limited sets of experimentally practical stimuli, units within deep nets can then be tested with vast and diverse stimuli to attempt to gain deeper understanding. For example, finding the downward-dog-paw and wolf-ear-steeple units raises the question of whether boundary curvature is encoded independent of other visual traits in V4 or in the CNN. Specifically, is it possible that V4 neurons that appear to encode curvature at a particular angular position are in fact also selective for texture or color features associated with a limited set of objects that have relevance to the monkey? Longer experimental sessions with richer stimulus sets will be required to test this in V4. Fourth, concrete, image-computable models can be used to address outstanding debates that may otherwise remain imprecise. For example, by visualizing the preferences of single units for natural stimuli after identifying and characterizing those units with artificial stimuli, our results speak to the debate on artificial vs. natural stimuli (*Rust and Movshon, 2005*) by showing that artificial stimuli are often able to reveal critical characteristics of the selectivity of units involved in complex mid-level (parts-based) to high-level (categorical) visual encoding, even when the visual dimensions of the artificial set explore only a minority of the feature space represented by the units. As another example, our results can help to address the debate of whether the visual system explicitly represents object boundaries (*Adelson and Bergen, 1991*; *Movshon and Simoncelli, 2014*; *Ziemba and Freeman, 2015*), which Movshon and Simoncelli describe as

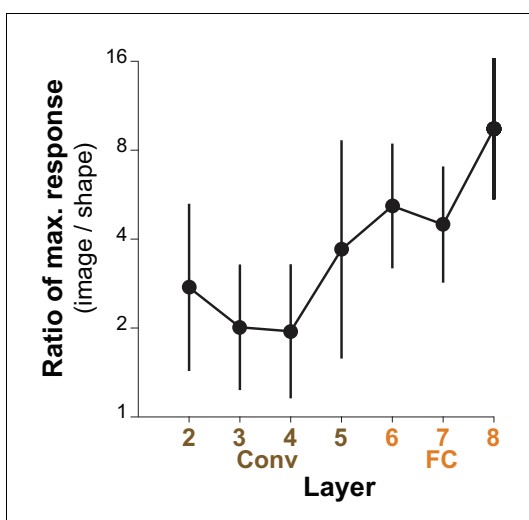

**Figure 13.** Comparing the maximum responses driven by images to those driven by shapes for APC-like units. For a given CNN unit, we computed the ratio of the maximum response across natural images (50,000 image test set) to the maximum response across our set of 362 shapes. The average of this ratio across the top ten APC-like units in each of seven layers (Conv2 to FC8) is plotted. Error bars show SD. In a few cases, the maximum response to shapes was a negative value and these cases were excluded: one unit for Conv3 and two for FC6 and FC7.

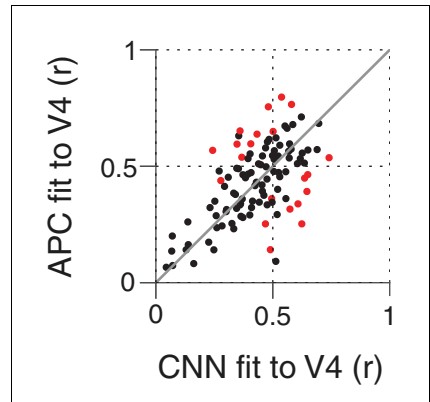

**Figure 14.** Comparing the ability of the APC model vs. single CNN units to fit V4 neuronal data. Showing r-values for cross-validated fits from both classes of model, black points correspond to V4 neurons for which neither model performed significantly better at predicting responses to the shape set. The APC model provided a better fit for red points above the line of equality, whereas points below the line correspond to neurons for which at least one unit within the trained CNN provided a better fit than any APC model.

follows: 'In brief, the concept is that the visual system is more concerned with the representation of the 'stuff' that lies between the edges, and less concerned with the edges themselves (*Adelson and Bergen, 1991*).' The models we have identified can now be used to pilot experimental tests of this rather complex, abstract idea.

Our approach also provides potentially valuable insight for machine learning. The connection between deep nets and actual neural circuits is often downplayed, but we found a close match at the level of specific single-unit selectivity. This opens the possibility that future studies could reveal more fine-scale similarities, that is matches of sub-types of single-unit selectivity, between artificial networks and the brain, and that such homology could become a basis for improving network performance. Second, translation invariance, seen as critical for robust visual representation, has not been systematically quantified for units within artificial networks. Determining why deeper layers in the network maintain a wide diversity of TI across units could be important for understanding how categorical representations are built. More generally, the art of characterizing units within complex systems using simple metrics and systematic stimulus sets, as practiced by electrophysiologists, can provide a useful way to interpret the representations learned in deep nets, thereby opening the black box to understand how learned representation contributes to performance.

### Further work

Our findings are consistent with the hypothesis that some CNN units share a representation of shape in common with V4 that is captured by the APC model. Examining whether these CNN units demonstrate additional V4 properties, beyond those examined here, would further test this hypothesis. For example, curvature-tuned V4 cells have been shown to (1) have some degree of scale invariance (*El-Shamayleh and Pasupathy, 2016*), (2) suppress the representation of accidental contours, for example those resulting from occlusion that are unrelated to object shape (*Bushnell et al., 2011*), (3) be robust against partial occlusions of certain portions of shape (*Kosai et al., 2014*), and (4) maintain selectivity across a spectrum of color (*Bushnell and Pasupathy, 2012*). Further studies like these are needed to more deeply probe whether the intermediate representation of shape and objects in the brain is similar to that in artificial networks. In addition to further study of functional response properties, it is important to understand how the network achieves these representations. For example, translation invariance was a key response property that allowed the trained network to achieve a V4-like representation, yet we are just beginning to understand what aspects of kernel weights, receptive field overlap, and convergence are critical to matching the physiological data. For CNNs to be valuable models of the nervous system, it will be important to understand what network properties support their ability to match representations observed in vivo.

## Materials and methods

### The convolutional neural network

We used an implementation of the well-known CNN referred to as 'AlexNet,' which is available from the Caffe deep learning framework (http://caffe.berkeleyvision.org; *Jia et al., 2014*). Its architecture (*Figure 2*) is purely feed forward: the input to each layer consists solely of the output from the previous layer. The network can be broken down into eight major layers (*Figure 2A*, left column), the first five of which are called convolutional layers (Conv1 through Conv5) because they contain linear spatial filters with local support that are repeatedly applied across the image. The last three layers are called fully connected (FC6 through FC8) because they receive input from all units in the previous layer. We next describe in detail the computations of the first major layer, which serves as a model to understand the later layers.

The first major convolutional layer consists of four sublayers (*Figure 2A*, orange, and *Figure 2C–F*, top four rows). The first sublayer, Conv1, consists of 96 distinct linear filters (shown in *Figure 1*) that are spatially localized to $11 \times 11$ pixel regions and that have a depth of three, corresponding to the red, green and blue (RGB) components of the input color images. The input images used for training and testing are $227 \times 227$ (spatial) x 3 (RGB) pixels. The output of a Conv1 unit is its linear filter output minus a bias value (a constant, not shown). Conv1 has a stride of 4, meaning that neighboring units have filters that are offset in space by four pixels. The output of each Conv1 unit is processed by a rectified linear unit in the second sublayer, Relu1, the output of which is simply the half-

wave rectified value of Conv1. These values are then pooled by units in the third sublayer, Pool1, which compute the maximum over a 3 × 3 pixel region (*Figure 2A*, gray triangles) with a stride of 2. The outputs of the Pool1 units are then normalized (see below) to become the outputs of units in the fourth sublayer, Norm1. These normalized outputs are the inputs to the Conv2 units in the second major layer, and so on. *Figure 2A* shows a scale diagram of the spatial convergence in the convolutional layers (major layers are color coded) along one spatial dimension. Starting at the top, the 11 × 11 pixel kernels (orange triangles) sample the image every four pixels, reducing the spatial representation to a 55 × 55 element grid (*Figure 2A*, column 4, lists spatial dimensions). The Pool1 layer reduces the representation to 27 × 27 because of its stride of 2. The Conv2 unit linear filters are 5 × 5 in space (red triangles) and are 48 deep (not depicted), where the depth refers to the number of unique kernels in the previous layer that provide inputs to the unit (see *Krizhevsky et al., 2012*), for details and their Figure 2 for a depiction of the 3D kernel structure).

These operations continue to process and subsample the representation until, after Pool5, there is a 6 × 6 spatial grid that is 256 kernels deep. Given the convergence between layers, the maximum possible receptive field (RF) size (i.e., extent along either the horizontal or vertical dimension) for units in each convolutional layer ranges from 11 to 163 pixels (*Figure 2B*) for Conv1 to Conv5, respectively. For example, the pyramid of support is shown for the central Conv5 unit (*Figure 2A*, dark blue triangle shows tip of upside-down pyramid), which has access to the region of width 163 pixels covered by Conv1 kernels (orange triangles). The receptive field sizes of units in the FC layers are unrestricted (not shown in *Figure 2B*). The last major layer, FC8, has a Prob8 sublayer that represents the final output in terms of the probability that the visual input contains each of 1000 different categories of object (e.g., Dalmation, Lampshade, etc.; see *Krizhevsky et al., 2012*, for details).

Units in the Norm1 and Norm2 sublayers carry out local response normalization by dividing their input value by a function (see *Krizhevsky et al., 2012*), their section 3.3) of the sum of squared responses to five consecutive kernels (indices from +2 to −2) along the axis of unique kernel indices (e.g., in Conv1, the indices go from 0 to 95 for the filters shown in *Figure 1*, from the upper left towards the right and down), thereby creating inhibition among kernels. *Figure 2D* (bottom row) lists the total number of units with unique kernels in each layer, and this defines the number of units that we examine here. In the Conv layers, we only test the units that lie at the central spatial position because they perform the same computation as their spatially offset counterparts. We analyzed a total of 22,096 units. To identify units for reproducibility in future studies, we refer to units by their layer name (e.g., Conv1) and a unit number, where the unit number is the index, starting at zero, within each sublayer and proceeding in the order defined in Caffe.

We tested the network in two states: untrained and fully trained. The untrained network has all weights (i.e., values within the convolutional kernels and input weights for FC layers) initialized to Gaussian random values with mean 0 and SD 0.01, except for FC6 and FC7 where SD = 0.005, and all bias values initialized to a constant of 0 (Conv1, Conv3, FC8) or 1 (Conv2, Conv4, Conv5, FC6, FC7). These initial bias values are relatively low to minimize the number of unresponsive units, which in turn guarantees a back propagation gradient for each unit during training. The fully trained network (available from Caffe) has been trained with stochastic gradient descent on large database of labeled images, ImageNet (*Deng et al., 2009*), with the target that the final sublayer, Prob8, has value 0 for all units except for a value of 1 for the unit corresponding to the category of the currently presented training image. To speed up training and mitigate overfitting, an elaborate training procedure was used that included a number of heuristics described in detail in *Krizhevsky et al. (2012)*.

## Visual stimuli

Our stimulus set (*Figure 3A*) is that used by *Pasupathy and Connor (2001)* to assess tuning for boundary curvature in V4 neurons. The set consists of 51 different simple closed shapes that are presented at up to eight rotations (fewer rotations for shapes with rotational symmetry), giving a total of 362 unique stimulus images. We rendered the shapes within a 227 by 227 pixel field with RGB values set to the same amplitude, thus creating an achromatic stimulus. The background value was 0, and the foreground amplitude was varied up to 255, the maximum luminance. This format matched the size and amplitude of the JPEG images on which the CNN was originally trained. The center of each shape was taken to be the centroid of all points on the finely sampled shape boundary. We fixed the foreground amplitude to 255 after varying it to lower values and finding that it made little

difference to the response levels through the network because of the normalization layers (see Results).

We set the size of our stimuli to be 32 pixels, meaning that the largest shape, the large circle (*Figure 3A* second shape from upper left), had a diameter of 32 pixels and all stimuli maintained the relative scaling shown in *Figure 3A*. This ensured the stimuli fit within the calculated RF of all layers except Conv1 with additional room for translations (see Maximum RF size, *Figure 2B*) and allowed all layers to be compared with respect to the same stimuli. We excluded Conv1 from our analysis because fitting the stimuli within the 11 by 11 pixel RFs would corrupt their boundary shape, would not allow room for testing translation invariance, and Conv1 is of less interest because of its simple function. In the V4 electrophysiological experiments of Pasupathy and Connor, stimuli were sized proportionally to each neuronal RF, as it can be difficult to drive a cell with stimuli that are much smaller than the RF. We tested sizes larger than 32 pixels (see Results) and found it did not substantially change our results.

## Electrophysiological data

For comparison to the deep network model, we re-analyzed data from two previous single-unit, extracellular studies of parafoveal V4 neurons in the awake, fixating rhesus monkey (Macaca mulattta). Data from the first study, *Pasupathy and Connor (2001)*, consists of the responses of 109 V4 neurons to the set of 362 shapes described above. There were typically 3–5 repeats of each stimulus, and we used the mean firing rate averaged across repeats and during the 500 ms stimulus presentation to constrain a model of tuning for boundary curvature in V4 (*Figure 3C*). To constrain translation invariance, we used data from a second study, *El-Shamayleh and Pasupathy (2016)*, because the first study used only two stimuli (one preferred and one antipreferred) to coarsely assess translation invariance. The data from the second study consists of responses of 39 neurons tested for translation invariance. The stimuli were the same types of shapes as the first study, but where the position of the stimuli within the RF was also varied. Each neuron was tested with up to 56 shapes (some of which are rotations of others) presented at 3–5 positions within the receptive field. Each unique combination of stimulus and RF position was presented for 5–16 repeats, and spike counts were averaged over the 300 ms stimulus presentation. Experimental protocols for both studies are described in detail in the original publications.

## Response sparsity

While many units in the CNN responded well to our shape set, there were also many units, particularly in the rectified (Relu) sublayers, that responded to very few or none of our shape stimuli. It was important to identify the very sparse responding units because they could bias our comparison between the CNN units and V4 neurons. We quantified response sparsity using the fourth moment, kurtosis (*Field, 1994*),

$$K = \frac{1}{n}\sum_i^n \frac{(x_i - \bar{x})^4}{\sigma^4},$$ (1)

where $x_i$ is the response to the $i^{\text{th}}$ stimulus, $n$ is the number of stimuli, and $\bar{x}$ and $\sigma$ are the mean and SD of the response across stimuli. This metric works for both non-negative and signed random variables, thus covering the outputs of all layers of the CNN. We excluded CNN units where response sparsity was outside the range observed in V4: 2.9 to 42 (*Figure 4—figure supplement 1*; see Results). We also found that such units gave degenerate fits to the APC model.

## Placing stimuli in the classical receptive field

In keeping with neurophysiology, we defined the classical receptive field (CRF) of a CNN unit as the region of the input from which our stimuli can elicit a response different from baseline, where baseline is defined as the response to the background input (all zeros). For example, to determine the horizontal extent of the CRF of a unit, we started with our stimulus set centered (in x and y) on the spatial location of the unit and determined whether there was a driven response (deviation from baseline) to any stimulus. We then moved the stimulus set left and right to cover a 100 pixel span in two pixel increments to find the longest set of contiguous points from which any response was elicited at each point. In other words, stimuli were centered on pixels ranging from 64 to 164 in the

227 pixel wide image. To account for the finite width of the stimuli, we subtracted the maximum stimulus width from the length of the contiguous response region and added one to arrive at the estimated extent of the CRF in pixels along the horizontal axis. Any unit with a CRF wide enough to contain three 2-pixel translations of our stimulus set was included in our analyses. Generally, this provided a conservative estimate of the receptive field, because most stimuli were narrower than the maximal-width stimulus, as observed in *Figure 3A*.

All analyses and plots of responses to translated shapes were made with respect to horizontal shifts of our vertically centered shape set. To verify that our conclusions did not depend on testing only horizontal shifts, we recalculated our metrics for vertical shifts and found them to be strongly correlated with those for horizontal shifts (*Figure 7—figure supplement 1*).

## The APC model

Our study focuses on the ability of CNN units to display a particular physiological property of V4 neurons—tuning for boundary conformation—which has been modeled using the angular position and curvature (APC) model introduced by *Pasupathy and Connor (2001)*. Conceptually, APC tuning refers to the ability of a neuron to respond selectively to simple shape stimuli that have a boundary curvature feature (a convexity or concavity) at a particular angular position with respect to the center of the shape. Unlike the CNN, the APC model does not operate on raw image pixel values, but instead on the carefully parameterized curvature and angular position of diagnostic elements of the boundaries of simple closed shapes (see example shape, *Figure 3B*). Each boundary element along the border of a shape can be mapped to a point in a plane heretofore referred to as the APC plane (*Figure 3C*). The responses, $R_i$, of a unit to the $i^{\text{th}}$ shape is given by:

$$R_i = k \, \max_j \left[ \exp\left( \frac{-(c_{i,j} - \mu_c)^2}{2\sigma_c^2} \right) \exp\left( \frac{-(a_{i,j} - \mu_a)^2}{2\sigma_a^2} \right) \right], \tag{2}$$

where the expression inside the square brackets is the product of two Gaussian tuning curves, one for curvature with mean $\mu_c$ and SD $\sigma_c$, and one for angular position with mean $\mu_a$ and SD $\sigma_a$. The curvature axis extends from $-1$ (sharp concavity) to $+1$ (sharp convexity), and the angular position is defined with respect to the center of the shape. The $j^{\text{th}}$ curvature value of the $i^{\text{th}}$ shape is encoded as $c_{i,j}$ and the angular position of that curvature element is $a_{i,j}$. The factor $k$ is a scaling constant. The max over these boundary elements is taken, thus the response depends only on the most preferred feature. In the original study (*Pasupathy and Connor, 2001*), these parameters were fit using a gradient descent method, the Gauss-Newton algorithm, from a grid of starting points across the APC plane. We instead discretely sampled the parameter space taking the Cartesian product of 16 values of $\mu_c$, $\sigma_c$, $\mu_a$ and $\sigma_a$, where the means were linearly spaced, the SDs were logarithmically spaced, and the end-points were set to match the range of values observed for the V4 cells when fit by the original Gauss-Newton method ($\mu_c \in [-0.5, 1]$, $\sigma_c \in [0.01, 0.98]$, $\mu_a \in [0°, 338°]$ and $\sigma_a \in [23°, 171°]$). We defined the best-fit model to be that which maximized Pearson's correlation coefficient between observed and predicted responses. We then found $k$ using a least squares fit. We found this to be more rapid, and the median correlation of the original V4 neurons to be the same to two decimal places as the Gauss-Newton fits (0.48), and had the assurance that the same models were tested on all units. We used Pearson's correlation coefficient two-tailed p-value to test for significance.

## Measuring translation invariance

To visualize translation invariance we created position-correlation functions by plotting the r-value of responses between a reference and an offset location as a function of distance (e.g., *Figure 6B and E–J*, red). To compare the fall-off in correlation to the fall-off in the RF profile (e.g., *Figure 6E–J*, green) of the CNN unit, we computed an aggregate firing rate metric—the square root of the sum of the squared responses across the stimulus set at each spatial position. For CNN units, this was used rather than the mean firing rate because responses could be positive or negative.

To quantify translation invariance in neuronal and CNN unit responses, we defined a metric, TI, that can be thought of as a generalization of the correlation coefficient. The correlation coefficient,

$$r = \frac{\mathrm{Cov}(\vec{p_1}, \vec{p_2})}{\mathrm{SD}(\vec{p_1})\,\mathrm{SD}(\vec{p_2})}, \tag{3}$$

which is bounded between $-1$ and $1$, measures how similar the response pattern is across two locations, where $\vec{p_1}$ and $\vec{p_2}$ are vectors containing the responses to all stimuli at positions 1 and 2. Our TI metric is,

$$\mathrm{TI} = \frac{\sum_{i \neq j} \mathrm{Cov}(\vec{p_i}, \vec{p_j})}{\sum_{i \neq j} \mathrm{SD}(\vec{p_i})\,\mathrm{SD}(\vec{p_j})}, \tag{4}$$

where the sums are taken over all unique pairs of locations, and $\vec{p_i}$ is the mean-subtracted column of responses at the $i^{\mathrm{th}}$ RF position. The numerator is the sum of the non-diagonal entries in the covariance matrix of the responses, and the denominator is the sum of the products of each corresponding pair of SDs. Thus, this metric is also bounded to lie between $-1$ and $1$, but it has an advantage over the average r-value across all unique pairs of locations because the latter would weight the r-value from RF locations with very weak responses just the same as those with very strong responses. For a simple model of neuronal translation invariance in which the variations of responses are described as the product of a receptive field profile and a shape selectivity function, our TI metric would take its maximum possible value, 1. If responses at all positions were uncorrelated, it would be 0.

We also evaluated an alternative metric, the separability index (*Mazer et al., 2002*; *Hinkle and Connor, 2002*) based on the singular value decomposition of the response matrix, but we found that it was biased to report higher translation invariance values for response matrices that reflected tuning that was more confined in space (i.e., smaller RF sizes) or more limited to a small range of shapes (i.e., higher shape selectivity). According to our simulations, our TI metric has the benefit of being unbiased with respect to receptive field size or selectivity of our response matrices, thereby facilitating comparisons across layers and diverse response distributions.

In testing the CNN, we finely sampled horizontal shifts of the stimulus set, as described above in 'Placing stimuli in the CRF'. The TI metric for any neuron was computed only for the set of contiguous locations for which the entire shape set was within the RF of the unit.

## Comparing CNN and APC model fits to V4 data

We examined whether the CNN units might directly provide a better fit to the V4 neural responses than does the APC model. This required us to compare, for each of the 109 V4 units, the best-fit unit in the pool of CNN units to the best fit provided by the APC model. In the case of the CNN, there are 22,096 units to consider (*Figure 2D*). In the case of the APC model, there are five parameters (see 'The APC model' above). We employed cross-validation to ensure that any differences in fit quality were not the result of one fitting procedure being more flexible than the other. In particular, we performed 50 fits on a random subset of 4/5 of the neural data, then measured the correlation of the fit model on the remaining 1/5. We took the mean of these 50 fits for each unit to be the estimate of test correlation, and the 95$^{\mathrm{th}}$ percentiles of the distribution of fits for identifying cells that deviate in their fit quality between two models (e.g., APC model and the CNN). To judge whether the variance explained by the CNN was largely distinct from that explained by the APC model we fit a V4 neurons best-fit CNN model to the residual of the fit of the APC model to a V4 neuron. If the correlation of the CNN unit to the V4 neuron remains high then the APC model and CNN explain different features of the response of the V4 neuron.

## Estimating the effect of the stochastic nature of neuronal responses

AlexNet produces deterministic, noise-free responses, whereas the responses of V4 neurons are stochastic. This raises the possibility that our conclusions might have been different if more trials of V4 data had been collected to reduce the noise in the estimates of the mean neuronal responses. In particular, trial-to-trial variability will tend to lower the correlation coefficient (r-value) between model and data.

To address this, we used the methods of *Haefner et al. (2009)* to remove the downward bias that trial-to-trial variability imparts on the r-value for our fits of the APC model to neuronal data. The method of Haefner and Cumming assumes that the neural responses have been appropriately

transformed to have equal variance across stimuli and that the averaged responses for each stimulus are normally distributed. For the case where the variance-to-mean relationship is, $\sigma^2(\lambda) = a\lambda$, where $\lambda$ is the mean response and $a$ is a constant (i.e., Fano factor is constant across firing rates), an often used transformation is the square root of the responses. Empirically, we have found that this transformation works well even when neural responses have a quadratic variance-to-mean relationship. After taking the square root of the responses, we estimated sample variance for each stimulus across trials and then averaged across stimuli to get $\bar{s}^2$. We made a least-squares fit of the model to the centered mean responses (grand mean subtracted from the mean for each stimulus). We then calculated the corrected estimate of explained variance:

$$\hat{R}_c^2 = \frac{\hat{\beta}^2 - \frac{\bar{s}^2}{n}}{\hat{\alpha}^2 + \hat{\beta}^2 - \frac{\bar{s}^2}{n}(m-1)}, \tag{5}$$

where $\hat{\beta}^2$ is the sum of squares of the model predictions (explained variance), $\hat{\alpha}^2$ is the sum of squares of the residuals from the model (unexplained variance), $\bar{s}^2$ is sample variance across trials, averaged for all stimuli, $m$ is the number of stimuli, and $n$ is the number of trials.

We used a different approach to estimate how much our TI metric for V4 neurons might be degraded by noise because TI is not a correlation coefficient and does not lend itself to the methods described above. In particular, for each V4 neuron tested with stimuli at multiple positions, we built an ideal model with perfect TI by taking the responses at the position that produced the greatest response and replicating them at the other positions, but scaling them to match the original mean at each RF position. We then used this set of sample means, which has TI = 1, to generate Poisson responses, simulating the original experiment 100 times and computing the TI value for each case. We took the average drop in TI (compared to 1) to be an estimate of the upper bound of how much the V4 neuron TI values could have been degraded by noise.

## Visualization

To visualize the features that drove a particular unit in the CNN to its highest and lowest response levels, we first ranked all images (or image patches) based on the response of the unit to the standard test set of 50,000 images for AlexNet. For units in the convolutional layers, we considered the responses at all x-y locations for a particular unique kernel. Thus, we found not just the optimal image, but also the optimal patch within the image that drove the kernel being examined. We then performed a visualization technique on the 10 most excitatory images and on the 10 most suppressive images. We followed the methods of *Zeiler and Fergus (2013)*, and used a deconvnet to project the response of the unit onto successive layers until we reached the input image. The deconvolved features can then be examined, as an RGB image, to provide a qualitative sense of what features within the image drove the unit to such a large positive or negative value.

## Acknowledgements

This work was funded by a National Science Foundation (NSF) Graduate Research Fellowship (DAP), a Google Faculty Research Award (WB and AP), the National Science Foundation CRCNS Grant IIS-1309725 (WB and AP), National Institutes of Health (NIH) Grant R01 EY-018839 (AP), NIH Office of Research Infrastructure Programs Grant RR-00166 to the Washington National Primate Research Center (AP), and NIH Grant R01 EY-027023 (WB) We thank Yasmine El-Shamayleh for sharing V4 data. We thank Blaise Aguera y Arcas for helpful suggestions and advice.

## Additional information

### Funding

| Funder | Grant reference number | Author |
| --- | --- | --- |
| National Science Foundation | Graduate Research Fellowship | Dean A Pospisil |
| National Science Foundation | CRCNS Grant IIS-1309725 | Anitha Pasupathy |

| | | Wyeth Bair |
|---|---|---|
| Google | Google Faculty Research Award | Wyeth Bair |
| National Institutes of Health | Grant R01 EY-018839 | Anitha Pasupathy |
| NIH Office of Research Infra-structure Programs | Grant RR-00166 to the Washington National Primate Research Center | Anitha Pasupathy |

The funders had no role in study design, data collection and interpretation, or the decision to submit the work for publication.

### Author contributions

Dean A Pospisil, Conceptualization, Data curation, Software, Formal analysis, Funding acquisition, Investigation, Visualization, Methodology, Writing—original draft, Project administration, Writing—review and editing; Anitha Pasupathy, Conceptualization, Resources, Data curation, Supervision, Funding acquisition, Methodology, Writing—review and editing; Wyeth Bair, Conceptualization, Resources, Software, Formal analysis, Supervision, Funding acquisition, Investigation, Visualization, Methodology, Writing—original draft, Writing—review and editing

### Author ORCIDs

Dean A Pospisil (iD) http://orcid.org/0000-0002-5793-2517
Anitha Pasupathy (iD) http://orcid.org/0000-0003-3808-8063

### Ethics

Animal experimentation: All animal procedures for this study, including implants, surgeries and behavioral training, conformed to NIH and USDA guidelines and were performed under an institutionally approved protocol at the Johns Hopkins University (Pasupathy and Connor, 2001) protocol #PR98A63 and the University of Washington (El-Shamayleh and Pasupathy, 2016) UW protocol #4133-01.

### Decision letter and Author response

Decision letter https://doi.org/10.7554/eLife.38242.sa1
Author response https://doi.org/10.7554/eLife.38242.sa2

## Additional files

### Supplementary files

• Transparent reporting form

### Data availability

No new datasets were generated in the course of this research. The model this research is based on is openly available from the Berkeley Artificial Intelligence Lab.

The following previously published dataset was used:

| Author(s) | Year | Dataset title | Dataset URL | Database and Identifier |
|---|---|---|---|---|
| Yangqing J, Shelhamer E, Donahue J, Karayev S, Long J, Girshick R, Guadarrama S, Darrell T | 2014 | BAIR BVLC CaffeNet Model 'reference caffenet' | http://dl.caffe.berkeleyvision.org/bvlc_reference_caffenet.caffemodel | Berkeley Vision and Learning Center, bvlc_reference_caffenet.caffemodel |

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
