## [Decision Letter]

[Editors’ note: the authors were asked to provide a plan for revisions before the editors issued a final decision. What follows is the editors’ letter requesting such plan.]

Thank you for sending your article entitled ""Artiphysiology" reveals V4-like shape tuning in a deep network trained for image classification" for peer review at *eLife*. Your article is being evaluated by two peer reviewers, and the evaluation has been overseen by a Reviewing Editor and Joshua Gold as the Senior Editor. The reviewers have opted to remain anonymous.

Given the list of essential revisions, including new experiments, the editors and reviewers invite you to respond within the next two weeks with an action plan and timetable for the completion of the additional work. We plan to share your responses with the reviewers and then issue a binding recommendation.

Summary:

The reviewers who read this paper had quite different views. In my role as the reviewing editor, here I integrate and summarize the different impressions and comments and provide the authors with some recommendations to revise the paper and best respond to the comments to improve the manuscript, with my hope that it will be possible to reconsider publication of this study.

This paper examines the hidden units of deep networks trained on image classification in order to determine whether their tuning properties are similar to those that have been reported previously in primate visual area V4. The authors probe a pre-trained deep network (AlexNet, used in many other computational neuroscience studies) with simple closed-contour binary shape stimuli. These stimuli were used in previous impactful neurophysiology studies from this group to probe contour curvature and angle tuning in area V4.

The paper reports three main general results. First, it shows that some units within the deep network can be well described by a model contour curvature model of V4 selectivity (the APC model) that was developed previously by the authors. Second, it shows that the deep network units well described by APC model also exhibit position invariance, similar to that observed in real V4 neurons and reported in several previous studies. Third, the authors use a network visualization method to show that deep network units well described by the APC model are predicted to be selective for a range of complex natural images and that the best natural images evoke larger responses from the APC units than are obtained using the simpler stimuli.

This is a well written and clear paper. The computational analyses appear to be solid. Treating a deep network trained on a natural image classification task as an object of synthetic neurophysiological investigation is an interesting enterprise, and it is gratifying to see the correspondence between units in the deep network and V4 neurons that are well fit by the APC model.

The "major concerns" section below provides details of the major question that we have faced, which is: What is the added value of this well-written paper? We see it in two almost opposite ways;

1) The paper has a significant and general value in highlighting the potential of advantages of computational models in interpreting neurophysiological data and cortical computations, as well as a potential of learning from neurophysiological data how to improve computational models.

2) The paper does not provide new insights. Instead, it is a reasonably good paper, oriented to a local community, that confirms the notion that hierarchical networks (either biological or artificial) end up representing similar hierarchical structure in natural images.

Essential revisions:

Most researchers agree that the critical questions regarding modeling of sensory system at large are the practical ones: (1) What stimulus should be used in neurophysiology data collection? (2) How much data should be collected? (3) How should models be validated? (4) What modeling framework should we use? The authors are invited to clarify and explain how the paper appropriately relate to these questions.

At some level, it seems like this study has to work out the way that it did. As several recent neurophysiology modeling studies using deep networks have argued, both deep networks and the primate visual system were "trained" (or evolved) using analyze natural images, and it is not surprising that both hierarchical networks end up representing similar hierarchical structure in natural images. From that point of view the results of the manuscript are unsurprising. The two major conclusions of the paper are that units in deep networks trained on classification and well fit by the APC model show selectivity for contour curvature and that they are locally positioned invariant. Both of these observations are completely consistent with an enormous amount of prior data. A wealth of prior research suggests that object borders are important for vision. Theoretical arguments suggest that object borders are a critical component of any viable representation of the objects in natural scenes, and many previous neurophysiology studies using both synthetic and natural images have shown that units in V2 and V4 are sensitive to object borders including curved borders. The demonstration that deep networks are positionally invariant is also not surprising. After all, the deep networks tested here are convolutional and therefore are designed to be locally positioned invariant. In fact, convolutional networks were inspired originally by the finding of position invariance in primate vision.

The important question, therefore, is not whether there is any correspondence between the tuning of visual neurons and the tuning of units in deep networks trained for image classification. Instead, the important questions concern the nature of this tuning, the distributional relationships between primate neurons and units in the deep networks and, ultimately, what that can tell us about the biophysical mechanisms underpinning the observed functional selectivity.

We would have been more enthusiastic if the analysis could reveal some new principles that could provide impetus to further experimental studies, or which could be used to help improve current models. For example, if the current deep network framework could support some fundamentally new approach to interpreting functional responses that would provide a foundation for building a mathematical model that can explain observed functional properties in terms of known biophysical building blocks present in cortical neurons, or if the deep networks could be used as the building blocks for such a model.

At this point, this is a well-done study that supports the current view that the primate visual system and deep networks trained on natural image classification both encode intermediate object structure such as contour curvature. However, this paper as written doesn't resolve controversies and doesn't provide information that would be useful for designing future experiments or models.

---

## [Author Response]

[Editors' note: the authors’ plan for revisions was approved and the authors made a formal revised submission.]

[…] The "major concerns" section below provides details of the major question that we have faced, which is: What is the added value of this well-written paper? We see it in two almost opposite ways;1) The paper has a significant and general value in highlighting the potential of advantages of computational models in interpreting neurophysiological data and cortical computations, as well as a potential of learning from neurophysiological data how to improve computational models.2) The paper does not provide new insights. Instead, it is a reasonably good paper, oriented to a local community, that confirms the notion that hierarchical networks (either biological or artificial) end up representing similar hierarchical structure in natural images.

Below we provide arguments, now highlighted in our manuscript, against three ideas articulated above: i) that our paper “does not provide new insights”, ii) that it is merely confirmatory, and iii) that it targets a local community.

i) Our manuscripts offers the following novel insights, highlighted in our revised Discussion:

1) We show that there are one-to-one correspondences between AlexNet model units and single V4 neurons in terms of tuning for translation invariant boundary curvature [Discussion, first paragraph]. This is despite the remarkable differences between the two systems in terms of architecture (e.g., lack of feedback in AlexNet), properties of individual units (e.g., no dedicated inhibitory neurons in AlexNet), training (limited visual environment of colony raised animal) and behavioral tasks (no motor output requirements for AlexNet) [Discussion, second paragraph]. This correspondence at the level of single units was unexpected and was not previously pursued in past studies (e.g., Yamins et al., 2014). [Discussion subsection “Comparison to previous work”, first paragraph]

2) Our results suggest that comparing properties of single units in AlexNet with cortical neurons can be a powerful tool to clarify complex processing beyond the earliest stages in primate cortex. We now emphasize that single-neuronal selectivity described by electrophysiologists might be questioned, but can gain more traction once similar selectivity is seen within an artificial network trained on a large bank of naturalistic inputs [Discussion subsection “Value of artiphysiology”]. Our paper highlights a general way forward for systems Neuroscience research.

3) Despite the CNN’s apparent complexity, form selectivity of some units in AlexNet can be described by a simple 5-parameter model. This is a novel strategy that complements the traditional approach of unit visualization, helping to defeat the notion of CNNs as mysterious black boxes. [Discussion subsection “Comparison to previous work”]

4) Our work shows that single units of AlexNet achieve V4-like translation invariance only after extensive training (our Figure 8B and Discussion ‘Training and translation invariance’). This will help to dispel the expectation that units in deep networks are translation invariant by design. Our paper can move the field beyond this misconception so that investigation can begin in earnest to understand how units within complex networks including the brain achieve translation invariance, we now emphasize this [Discussion subsection “Training and translation invariance”].

5) Our results speak to the artificial vs. natural stimulus debate. They reveal that artificial stimuli can be quite diagnostic even in systems trained only on natural stimuli and even in deep layers of the system where the artificial stimuli may drive only a small fraction of the response dynamic range. [Discussion subsection “Value of artiphysiology”]

6) Our study identifies the first image-computable, biologically-plausible circuit model that matches quantitatively the boundary shape tuning and TI in V4 [Discussion, first paragraph]. We thus provide a concrete, openly available tool to test complex theories of image statistic vs. object centered encoding in V4 (Ziemba and Freeman, 2014).

ii) Our results are not merely confirmatory: there is little consensus about processing in area V4 and there are major differences between the primate brain and CNNs in terms of architecture and training. Thus, there is not a solid basis to predict, at the single-unit level, the outcome of a comparison of form selectivity between deep nets and area V4.

1) There is currently no accepted model among electrophysiologists of what V4 does; therefore, to say that one knows in advance that a particular single-neuron selectivity in V4 (here, selectivity for boundary shape of simple forms) will emerge from a given artificial network is not tenable. Neuroscientists do not yet agree on what V4 neurons do, and different studies use widely different experimental paradigms and propose different circuit models that make different and sometimes directly contradictory predictions. This fact is demonstrated nicely by the current review, which questions whether the APC model is really any good at describing important properties of V4 neurons, and whether it is even valid to use in our study. This reflects the reality that we face regularly, and it is based on disagreements in published papers, public talks and anonymous critique that demonstrate the overall feeling of doubt that anybody understands what is going on in an area like V4. This view conflicts with the idea that, “we already know what features are represented in single units in the brain and in units in deep nets (and we know these are the same)”. So which is it: do we already know all of this, or are we confused about basic V4 selectivity? Because of this lack of clarity about V4, we believe that our paper will be of great interest and spur important debate in the community. This highlights an important insight of our paper: that single-neuronal selectivity described by electrophysiologists might seem questionable, but can gain more traction once it is seen that such selectivity can arise within an artificial network trained on a large bank of naturalistic inputs, we now discuss this point in our revised manuscript [Discussion subsection “Value of artiphysiology”]. Importantly, this is not local to V4, the circuits and roles of neurons in V2, V3, IT, and beyond the visual cortex, are not agreed upon either. Thus, our paper highlights a general way forward in neuroscience.

2) There is wide disagreement about the validity of deep convolutional networks as robust models of human visual function. Many see these models as using tricks of image statistics to achieve good results – for example, journal articles highlight adversarial images, where deep nets fall down, and cases where artificial networks can fail spectacularly in challenging situations that are trivial for the primate visual system. The argument has been made that the primate brain depends on feedback and structural knowledge of the world, and these elements are not yet properly incorporated into the design and training of artificial networks. Primates have different training inputs and different tasks compared to the type of deep net studied here: they manipulate objects and experience them in 3D and over time, whereas AlexNet does not; we now include this point in our manuscript [Discussion, second paragraph]. We and others have considered that border ownership and boundary shape selectivity are critical for organisms that must form their hands to grasp objects with great certainty, or perish. It remains uncertain going forward to what extent we should expect specific representations at the single unit level to align across such diverse systems. This argues against the predictability of what types of specific neuronal selectivity might be shared between the brain and deep nets.

3) There is wide agreement that deep nets are difficult to understand and provide little insight in spite of their amazing performance on specific tasks. This contradicts the notion that one already knows what to expect when examining the internal representation of single units in the deep net. Here we show that a simple, insightful 5-parameter model can be related to parts of a deep net [Discussion subsection “Comparison to previous work”]. Previous studies that have compared responses have not gone down to the single unit level, and thus have left the mystery of the black box intact.

4) If another study comes out next month where it is found that a major property of V4 neurons (or of neurons in V2 or IT) is *not* present in the deep net (e.g., blur tuning, accidental contour discounting, color-invariant shape tuning, etc.), can one not equally well then say, “Of course this is not surprising because deep nets and the brain are so fundamentally different: different architecture, different sensory inputs and different tasks?” The fact that the opposite view (to that in Summary #2 above) could also be backed up by a large set of viable arguments suggests that our results cannot be taken as known in advance.

5) A phrase is repeated in the summary and below, “that hierarchical networks (either biological or artificial) end up representing similar hierarchical structure in natural images.” To argue that this obviates our results would imply that one has accepted the idea that the visual system is best described as hierarchical and that we know what features are represented therein. However, this is not the case. While there is general consensus that boundaries (“edges”) and textures in images are critical, it is still unknown what features are encoded beyond V1, how visual receptive fields are built, how selectivity actually arises in a circuit, and how best to conceptualize visual processing. There is still deep debate as to the function of feedback, and it is actively investigated whether predictive coding is a better model for the cortex at many levels. Visual neuroscientists work in a world where currently very little is established and agreed upon about circuitry and single neurons in mid-level processing. The phrase quoted above about hierarchy presents a broad, classical notion that has been a useful point of reference but has remained under debate for fifty years. Currently, vast amounts of funding and scientific effort are aimed at trying to understand cortical circuitry and function. This argues against the ideas that (a) we already know the ventral stream is a hierarchy like a feedforward deep net, (b) we know which sensory features are represented throughout, and (c) we have already uncovered image-computable models to explain visual representation that aligns with that in neuronal circuitry.

iii) Our paper does not merely target a local community. We believe that our paper takes a novel approach in connecting two substantial communities – electrophysiologists who depend upon simple and limited stimulus paradigms and the machine learning community that aims to improve intelligent performance in artificial systems. Only recently have electrophysiologists had access to artificial networks that rival their physiological model systems in terms of having daunting complexity and strikingly good “behavioral” performance. Our results suggest to electrophysiologists that they may be able to harness deep nets for single-unit level comparisons in terms of neural representation. To the machine learning community, it reinforces the notion that the brain remains an important system for comparison and guidance, and that the craft of interpreting the innards of black boxes, honed by electrophysiologists, might be usefully applied to make sense of internal representation in deep nets. Our paper should have broad interest beyond local communities because it goes toward the question of the degree to which man-made intelligent systems can end up looking like the brain at a fine scale. We believe our paper both connects and transcends local communities. We now emphasize this starting with the opening line of our Abstract and by making specific sections to explain what electrophysiologists can learn and what the machine learning community can learn [Discussion subsection “Value of artiphysiology”].

Essential revisions:Most researchers agree that the critical questions regarding modeling of sensory system at large are the practical ones: (1) What stimulus should be used in neurophysiology data collection? (2) How much data should be collected? (3) How should models be validated? (4) What modeling framework should we use? The authors are invited to clarify and explain how the paper appropriately relate to these questions.

We highlight below how our paper relates to these four questions and will emphasize the main points within our revised manuscript. Due to the open-ended, general nature of this invitation, we attempt to keep our replies brief to focus more on specific criticisms further below.

1) What stimuli to use? A well-known debate in neurophysiology is whether to use artificial or natural stimuli (e.g., Rust and Movshon, 2005). Our artificial shape stimuli were substantially different from the training images for AlexNet, thus one could expect there to be little relationship between responses of the units to shapes and responses to natural images. We found a close relationship in many cases where the shape tuning of the units was reflective of their shape tuning over natural stimuli (Figure 11-12). Even for units at the deepest level of the network, the artificial electrophysiological stimuli were able to correlate strongly with the output category. There were also cases where characterization with the artificial shapes did not appear to relate to that with natural images, and this provides concrete examples for further research to gain insight into the determining factors [Discussion subsection “Visualization of V4-like CNN units”].

More importantly, by identifying an image-computable model for V4 boundary-form selective units, we offer the community an opportunity to test and optimize their stimuli on a model that is better than any V4 form-selective model as far as we know [Discussion subsection “Comparison to previous work”]. By showing not only which stimuli best drive these units, but also which most suppress them, we provide completely novel insight into stimulus design to test opponency and inhibition in mid-level form processing. We are using these novel CNN-unit models to design stimulus selection algorithms in our electrophysiological lab, and have just collected data from our first V4 units using this method. Such methods are generally of interest to others (e.g., Cowley et al., 2017, NIPS).

These models are also useful for developing stimulus sets to quantitatively understand form vs. texture tuning at the level of single units and circuits, and can be used to operationalize the difference between the notion of selectivity for things vs. stuff (Adelson, 1982), e.g., for object boundaries vs. general image statistics [Discussion Subsection “Value of artiphysiology”].

2) How much data to collect ? Our paper relates to this in several ways.

A) We show that TI can be estimated along one axis of translation (horizontal) and this metric has strong correlation along the other axis (vertical), and with 2D translation [Results subsection “Translation Invariance”, last paragraph]. This is important for studies that try to understand TI in the cortex (another debated area: e.g., Sharpee et al., 2013; Rust and DiCarlo, 2010), where stimulus sets may need to be large to have ample diversity but then cannot be repeated in their entirety across a dense 2D grid.

B) Our simulations and noise-correction procedures [Materials and methods subsection “Estimating the effect of the stochastic nature of neuronal responses”] show that with ~5 stimulus repeats, our R-values for APC model fits are under-estimated only by a small amount [now shown as pink line added into Figure 5G].

C) We provide novel V4 models as tools for the community to estimate when they have collected enough different stimulus dimensions to have fully understood a unit in a complex, non-linear network.

3) How should models be validated? We add substantively to a larger discussion on model validation. Prior studies on the effectiveness of deep neural networks for modeling the nervous system have not provided insight at the level of single neurons and electrophysiological selectivity (see for example Yamins et al., 2014; Khaligh-Razavi and Kriegeskorte, 2014). Their approaches emphasized explained variance for population-level fits. Here we take a different approach by choosing specific known response properties of single units (translation invariance and shape selectivity) and determine whether units in an artificial net achieve these actual neural response properties. Furthermore, we use visualization of unit responses to natural images to qualitatively validate whether the representation that these response properties are intended to capture (an object-centered representation of boundary) does in fact hold across natural images. We believe this level of model validation, which includes quantitative and conceptual level agreement to documented neuronal selectivity, pushes the field significantly beyond what has been done, we now state this in the revised manuscript [Discussion subsection “Comparison to previous work”]. Our results allow modelers to focus on specific neural selectivities and work with concrete, identified circuits that have biologically plausible components which we emphasize now. Furthermore, the units in AlexNet that we point to are publicly available, image-computable models, thus allowing maximal transparency for others to validate or invalidate the models by any method they choose.

4) What modeling framework to use? We have used two fundamentally different modeling frameworks in this study: the APC model, a simple descriptive model, and AlexNet, an image-computable statistical machine-learning model, and from this we have identified elements that can be extracted and analyzed to drive modeling at the circuit and biophysically plausible level. We now clarify in the Discussion [second paragraph in subsection “Comparison to previous work”] how using modeling frameworks at different levels of complexity (the 5-parameter APC model vs. the elaborate CNN) generates insight. We discuss how the combination of modeling frameworks is crucial, where an image computable model sheds light on potential circuit-level implementation, while the functional APC model captures the abstract representations (e.g., object-centeredness) theorized to be encoded and provides broad interpretation of neural responses. Because AlexNet is image-computable and feedforward, it allows one to extract and analyze the modular upstream circuitry driving identified units. These excerpts can then be compared to the best attempts at hand-made circuits (e.g., H-Max and many other cortical models of visual neuroscientists), refined to reach any desired level of biological plausibility, and analyzed to extract novel general principles. We believe this will be an important way forward in understanding form selectivity and translation invariance in the visual system.

At some level, it seems like this study has to work out the way that it did. As several recent neurophysiology modeling studies using deep networks have argued, both deep networks and the primate visual system were "trained" (or evolved) using analyze natural images, and it is not surprising that both hierarchical networks end up representing similar hierarchical structure in natural images. From that point of view the results of the manuscript are unsurprising.

For general arguments as to why our results did not have to work out the way they did, see our replies to the Summary above. To this more specific point, it is rational to expect that similar systems trained on similar inputs for similar tasks could achieve similarity in representation. But, before deciding whether CNNs and the macaques in this study can be considered sufficiently similar systems, several specific questions would need to be answered. Namely,

1) how similar is the macaque to AlexNet in terms of simple visual input? The macaques in the study were raised in an animal colony, and thus in a limited visual environment. They never saw ImageNet images and probably never saw even a single instance of the overwhelming majority of the 1000 image categories of AlexNet. They did not see the forest, the ocean, the sky nor other important contexts for AlexNet categories. These animals were never exposed to natural images as part of an experiment or training. We are aware of no scientific study that estimates how close the input statistics are for such macaques and AlexNet, and it would be an oversimplification to say there was one set of natural scene statistics that caused all visual systems to develop the same way (there is great diversity in biological visual systems).

2) How similar are the training signals? The macaque visual system could be shaped by the need to physically interact in real time in a 3D dynamic world. It is unknown how much influence this might have on representation. AlexNet does not interact with the world that generated the images that it senses, nor is it even given information about the location of object boundaries in its images for the categorization task for which it was trained.

3) How do the vastly different architectures of the macaque and AlexNet alter single-unit representation? AlexNet does not have a retina or LGN nor feedback from higher areas, nor dedicated excitatory and inhibitory neurons, nor does it have to compute with action potentials. It is unknown how these elements influence single-unit representation in an appropriate context.

4) Will a small set of artificial stimuli from the electrophysiology lab be sufficient to probe a network that was never trained on such impoverished and noise-free inputs? Overall, without knowing in advance a lot more about the answers to such questions, we do not see how one could scientifically predict our results about the similarity of mid-level single-unit representations across these two very different systems. We discuss these points in the revised manuscript [Discussion second paragraph].

For example, Yamins et al. (2014) in part motivate the regression of many units’ responses in the CNN onto multi-unit data in cortex by arguing it would be unlikely to see one-to-one correspondences between CNN units and neurons in the brain. This argument assumed a highly distributed representation where it would be unlikely for two axes (single units) of this representation to align but only by looking at a population could a shared representation be discovered. Our results add to this debate by opening the possibility that a match could be made at the level of single unit selectivity.

The two major conclusions of the paper are that units in deep networks trained on classification and well fit by the APC model show selectivity for contour curvature and that they are locally positioned invariant. Both of these observations are completely consistent with an enormous amount of prior data.

To clarify, we show that many units can fit the boundary curvature (APC) model for stimuli centered in the RF. But not all of these units are locally translation invariant. When the additional criterion of translation invariance (TI) is added, then fewer units are good fits to the APC/TI model (relative to V4 selectivity). When visualization and response dynamic range are also considered, only a minority of units appear to match the APC/TI model quantitatively and conceptually. We added additional clarification in the text that this match only occurs for some units in the middle layers [Discussion, first paragraph].

No study has ever fit a quantitative model of boundary curvature tuning to units in AlexNet, and no study has ever systematically estimated TI for units as we have done (using a systematic set of stimuli like those in electrophysiology and with a metric that controls for RF size and other factors). Studies of translation invariance in these nets have had mixed and conflicting results, as we report in the Discussion [subsection “Other studies of TI in CNNs”]. Deep net studies disagree about TI across layers, and V4 studies disagree about TI in neurons. Thus, our observations are not ‘completely consistent with an enormous amount of prior data’ as far as the published literature (see below for further details). Nevertheless, our results are consistent with some conclusions of previous work, and because all correct studies should be consistent with each other, this is not a shortcoming of our work. Overall, there is no other study in the category of our study – comparing single unit selectivity for mid-level vision in the macaque and a CNN [Discussion subsection “Comparison to previous work”].

A wealth of prior research suggests that object borders are important for vision. Theoretical arguments suggest that object borders are a critical component of any viable representation of the objects in natural scenes, and many previous neurophysiology studies using both synthetic and natural images have shown that units in V2 and V4 are sensitive to object borders including curved borders.

We generally agree with these statements and they are not criticisms, but premises, of our work aiming to find image-computable models to advance our understanding of the primate visual system. In this context it is interesting to note ideas of Movshon and Simoncelli (2014) and Ziemba and Freeman (2015), who argue that the whole demonstration of border selectivity could just be a result of selectivity for higher order image statistics. Thus, there is still disagreement about how such basic features are processed in the cortex, and having deep nets as useful working models could add substantially to this debate and could ultimately resolve this debate. We have revised our manuscript to emphasize these points [Discussion subsection “Value of artiphysiology”].

The demonstration that deep networks are positionally invariant is also not surprising. After all, the deep networks tested here are convolutional and therefore are designed to be locally positioned invariant. In fact, convolutional networks were inspired originally by the finding of position invariance in primate vision.

While it is a commonly held belief that deep networks are translation invariant by design, we find that they only achieve actual translation invariance in their units after extensive training (see Figure 7B and Discussion ‘Training and translation invariance’). It is important that this commonly held belief is dispelled so that investigation can begin in earnest to understand how units within complex networks including the brain achieve translation invariance. We now emphasize this point in our revised manuscript [Discussion subsection “Other studies of TI in CNNs”].

Our results demonstrate that TI of single-unit form selectivity tends to increase with depth in the network but is highly variable from unit-to-unit even within a layer. We discuss in “Other studies of TI in CNN’s” how prior work published in the computer vision literature reports conflicting results as to the progression of invariance in layers of CNNs. We believe differences in reported TI across layer are the result of a lack of consistent definitions and absence of careful controls. Again this shows that the results we present are not simply ‘completely consistent with an enormous amount of prior data’ as far as the published literature. We appreciate being made aware of the study by Güçlü and van Gerven (2015) that used an invariance metric in a deep network. Their measure is fundamentally different from our TI–they fit a Gaussian surface to responses of units to translations of a single preferred stimulus, then take the median extent of the Gaussian across units in a layer as a measure of invariance for the layer. This metric is more similar to an electrophysiological estimate of average receptive field width across units in a layer; it does not measure the consistency of form selectivity to diverse stimuli across space. We now discuss their study in our revised manuscript [Discussion subsection “Other studies of TI in CNNs”]. Our TI metric was designed to do the latter, in keeping with notions from other electrophysiological studies.

The important question, therefore, is not whether there is any correspondence between the tuning of visual neurons and the tuning of units in deep networks trained for image classification. Instead, the important questions concern the nature of this tuning, the distributional relationships between primate neurons and units in the deep networks and, ultimately, what that can tell us about the biophysical mechanisms underpinning the observed functional selectivity.

We generally agree. Our study is focused on the nature of form tuning in V4 and in AlexNet. We test whether the nature of the encoding for form is similar in terms of a specific model of object-centered boundary curvature derived from electrophysiology. We ask whether, in the eyes of a scientist using common experimental paradigms, units in the network could be indistinguishable from neurons in the macaque brain, and we characterized the distribution of those units within and across layers in the network. We highlight specific units for further study, thus providing the entire community of experimentalists and modelers with the first set of specific, novel circuit models for a type of neuronal form selectivity (Table 1). Other studies have reported more general observations and have not looked at the nature of specific single-neuron representation, or have simply focused on other questions. Figure 5G and 7A in our manuscript report the distribution of curvature tuned and translation invariant units, respectively, across layers of AlexNet.

As for biophysical mechanisms, the operations in a deep neural network are generally biologically plausible (addition, multiplication, division, max-pooling) and have been used by neuroscientists attempting to hand-tune cortical models (e.g., HMax model and various normalization models). Because of this, the units that we have identified, and that are publicly available in a well-documented system (Caffe), can be used to make predictions about cortical circuits in terms of the number and types (of selectivity) of units that could be combined to achieve selectivity and invariance in V4. This can guide predictions about how features are linked together from V1 to V2 to V4. We are currently using the models identified in our paper to work backwards and forwards in the deep net to understand what computations are critical to achieving V4-like boundary shape selectivity. With respect to TI, we are just beginning to understand what properties of kernel weights, receptive field overlap, and convergence are critical to matching the physiological data. We describe these efforts in our manuscript. [Discussion subsection “Further work”]

We would have been more enthusiastic if the analysis could reveal some new principles that could provide impetus to further experimental studies, or which could be used to help improve current models. For example, if the current deep network framework could support some fundamentally new approach to interpreting functional responses that would provide a foundation for building a mathematical model that can explain observed functional properties in terms of known biophysical building blocks present in cortical neurons, or if the deep networks could be used as the building blocks for such a model.

Our paper does help to improve current models of V4. Specifically, we provide a novel set of models (i.e., clearly identified units and their supporting circuits in AlexNet) that are the best and only models for a mid-level, single-neuron, visual form selectivity that meets critical quantitative criterion used in electrophysiology. This opens up many lines of hypothesis testing, model refinement and stimulus selection. In terms of biophysical building blocks, synaptic weights, thresholds, pooling, inhibition and excitation (positive and negative weights), and normalization are plausible operations at the circuit and system level, and these can be fairly easily translated into more refined biologically plausible models (e.g., E-I spiking circuits). We now emphasize these points in the revised manuscript [Discussion, subsection “Comparison to previous work”].

At this point, this is a well-done study that supports the current view that the primate visual system and deep networks trained on natural image classification both encode intermediate object structure such as contour curvature. However, this paper as written doesn't resolve controversies and doesn't provide information that would be useful for designing future experiments or models.

Resolving controversies: Our paper provides novel insight bearing on several important controversies listed below. For a description of the value of our study in identifying novel, useful models and designing experiments, please see our replies above.

Natural vs. artificial stimuli: We show that artificial stimuli can be quite diagnostic even in systems only trained on natural stimuli and even in deep layers of the system where the artificial stimuli may drive only a small fraction of response dynamic range. We directly show that artificial and natural stimuli can give highly consistent impressions of mid-level shape selective units in a complex non-linear network. We now discuss how this paper contributes to this debate in the revised manuscript [Discussion subsection “Value of artiphysiology”].

Object boundaries vs. ‘stuff’: By identifying the first image-computable, biologically-plausible circuit model that matches quantitatively the boundary shape tuning and TI in V4, we provide a concrete, openly available tool to test complex theories of image statistic vs. representations of object boundaries. Movshon and Simoncelli (2014) describe this debate: “In brief, the concept is that the visual system is more concerned with the representation of the “stuff” that lies between the edges, and less concerned with the edges themselves (Adelson and Bergen, 1991).” We show how our paper contributes to this debate in the revised manuscript [Discussion subsection “Value of artiphysiology”.]

Is translation invariance built-in or learned?: As noted above, it is a widely held belief that translation invariance is built in to convolutional neural networks. Our work helps to move beyond this belief by demonstrating the importance of measuring TI for single units. We emphasize this important point in our revised manuscript [Discussion subsection “Training and translation invariance”].

Do deep nets develop representations like those in the brain?: Our paper is the first to show a single-unit level correspondence for mid-level visual form processing between deep nets and the brain [Discussion subsection “Comparison to previous work”].